# FAST ADVERSARIAL TRAINING AGAINST SPARSE ATTACKS REQUIRES LOSS SMOOTHING

## ABSTRACT

This paper studies fast adversarial training against sparse adversarial perturbations bounded by $l_0$ norm. We demonstrate the challenges of employing 1-step attacks on $l_0$ bounded perturbations for fast adversarial training, including degraded performance and the occurrence of catastrophic overfitting (CO). We highlight that CO in $l_0$ adversarial training is caused by sub-optimal perturbation locations of 1-step attack. Theoretical and empirical analyses reveal that the loss landscape of $l_0$ adversarial training is more craggy compared to its $l_\infty$, $l_2$ and $l_1$ counterparts. Moreover, we corroborate that the craggy loss landscape can aggravate CO. To address these issues, we propose Fast-LS-$l_0$ that incorporates soft labels and the trade-off loss function to smooth the adversarial loss landscape. Extensive experiments demonstrate our method can overcome the challenge of catastrophic overfitting, achieve state-of-the-art performance, and narrow down the performance gap between 1-step and multi-step adversarial training against sparse attacks.

## 1 INTRODUCTION

Deep neural networks have been shown vulnerable to adversarial perturbations (Szegedy et al., 2013). To achieve robust models, comprehensive evaluations (Athalye et al., 2018; Croce & Hein, 2020b; Croce et al., 2020) have demonstrated that adversarial training (Madry et al., 2018) and its variants (Croce & Hein, 2020a; Sehwag et al.; Rebuffi et al., 2021; Gowal et al., 2021; Rade & Moosavi-Dezfooli, 2021; Cui et al., 2023; Wang et al., 2023) are the most effective methods. However, adversarial training is generally computationally expensive because generating adversarial perturbations in each training step needs multiple forward and backward passes of the model. Such efficiency issues hinder the scalability of adversarial training to large models and large datasets.

Improving the efficiency of adversarial training is tricky. Some works (Shafahi et al., 2019; Zhang et al., 2019a; Wong et al.; Sriramanan et al., 2021) employ faster but weaker 1-step attacks to generate adversarial perturbations for training. However, such methods may suffer from *catastrophic overfitting (CO)* (Kang & Moosavi-Dezfooli, 2021): the model overfits these weak attacks instead of achieving true robustness against adaptive and stronger attacks.

On the other hand, most existing works (Madry et al., 2018; Tramer & Boneh, 2019; Jiang et al., 2023) focus on studying adversarial perturbations bounded by $l_\infty$, $l_2$ or $l_1$ norms. In these scenarios, the set of allowable perturbations is convex, which facilitates optimizing adversarial perturbations and thus adversarial training. However, there are many scenarios in real-world applications where sparse perturbations, bounded by the $l_0$ norm, need to be considered (Modas et al., 2019; Croce & Hein, 2019; Croce et al., 2022; Zhong et al., 2024). Since the $l_0$ norm is not a proper norm, the set of all allowable perturbations in this case is not convex. Consequently, from an optimization perspective, obtaining robust models against sparse perturbations becomes more challenging. Compared with the $l_\infty$, $l_2$ and $l_1$ counterparts, more steps are needed to generate strong $l_0$ bounded perturbations, making the corresponding adversarial training even more computationally expensive.

Among algorithms aiming at obtaining robust models against sparse perturbations, sAT and sTRADES (Zhong et al., 2024) stand out as the most effective ones. These methods employ adversarial training against Sparse-PGD (sPGD) (Zhong et al., 2024). However, they still require 20 steps to generate adversarial perturbations in each training step to achieve decent performance. As

Table 1: Robust accuracy of sAT and sTRADES (Zhong et al., 2024) with different steps ($t$). The evaluation is based on Sparse-AutoAttack (sAA) (Zhong et al., 2024), where the sparsity level is $\epsilon = 20$. The models are PreactResNet-18 (He et al., 2016a) trained on CIFAR-10 (Krizhevsky et al., 2009).

|  | sAT ($t = 1$) | sAT ($t = 20$) | sTRADES ($t = 1$) | sTRADES ($t = 20$) |
|---|---|---|---|---|
| Robust Accuracy | 0.0 | 36.2 | 31.0 | 61.7 |

demonstrated in Table 1, naively decreasing the number of steps to 1 leads to a significant performance decline for both sAT and sTRADES.

In this work, we investigate the challenges associated with fast adversarial training against sparse perturbations, including training instability caused by catastrophic overfitting (CO) and performance decline in both robust accuracy and clean accuracy. Specifically, we highlight that CO in $l_0$ adversarial training is caused by sub-optimal perturbation locations of 1-step attack. Our observation indicates that adjusting the perturbation magnitudes alone cannot help mitigate CO in this context, so existing CO mitigation methods (Kim et al., 2020; Andriushchenko & Flammarion, 2020; Zheng et al., 2019; Huang et al., 2023) used in other cases do not work in the $l_0$ scenario. Following, we provide empirical and theoretical evidence to illustrate that the loss landscape of adversarial training against $l_0$ bounded perturbations is notably more craggy compared to its $l_\infty$, $l_2$, and $l_1$ counterparts. Remarkably, these observations hold true even when only a single pixel is perturbed. Furthermore, we corroborate that the craggy loss landscape further aggravates CO in $l_0$ adversarial training.

Drawing from these insights, we propose to utilize soft labels and a trade-off loss objective function to enhance the smoothness of the adversarial loss objective function, thereby improving the performance of fast adversarial training against sparse perturbations. In addition to the performance, we showcase that these techniques can also eliminate CO, thus improving training stability. Finally, our extensive experiments demonstrate that smoothing the loss landscape can effectively narrow the performance gap between 1-step adversarial training and its multi-step counterparts.

To the best of our knowledge, this work is the first to investigate fast adversarial training in the context of $l_0$ bounded perturbations. We summarize the contributions of this paper as follows:

1. We highlight that catastrophic overfitting (CO) in fast $l_0$ adversarial training is caused by sub-optimal perturbation locations of 1-step attack. Popular techniques in fast $l_\infty$, $l_2$ and $l_1$ adversarial training are ineffective in the $l_0$ case.

2. We theoretically and empirically demonstrate that the adversarial loss landscape is more craggy in the $l_0$ cases than in other cases, which further aggravates CO in $l_0$ adversarial training. In this regard, we propose **Fast-LS-$l_0$** which incorporates the techniques of soft labels and the trade-off loss function to provably smooth the adversarial loss landscape.

3. Our comprehensive experiments demonstrate that smoothing the adversarial loss landscape greatly narrows the performance gap between 1-step $l_0$ adversarial training and its multi-step counterpart. Our method establishes a new state-of-the-art performance for efficient adversarial training against sparse perturbations.

**Notation and Terminology** We consider a classification model $F(\boldsymbol{x}, \boldsymbol{\theta}) = \{f_i(\boldsymbol{x}, \boldsymbol{\theta})\}_{i=0}^{K-1}$, where $\boldsymbol{x} \in \mathbb{R}^d$ is the input, $\boldsymbol{\theta}$ represents the parameters of the model, and $K$ is the number of classes, $f_i(\boldsymbol{x}, \boldsymbol{\theta})$ is the logit of the $i$-th class. Correspondingly, we use $\{h_i\}_{i=0}^{K-1}$ to represent the output probability of each class, which is the result of softmax function applied to $\{f_i\}_{i=0}^{K-1}$. Therefore, the loss objective function $\mathcal{L}$ based on the cross-entropy is calculated as follows:

$$\mathcal{L}(\boldsymbol{x}, \boldsymbol{\theta}) \stackrel{\text{def}}{=} -\sum_{i=0}^{K-1} y_i \log h_i(\boldsymbol{x}, \boldsymbol{\theta}) \stackrel{\text{def}}{=} -\sum_{i=0}^{K-1} y_i \log \frac{\exp(f_i(\boldsymbol{x}, \boldsymbol{\theta}))}{\sum_{j=0}^{K-1} \exp(f_j(\boldsymbol{x}, \boldsymbol{\theta}))} \tag{1}$$

where $\boldsymbol{y} = [y_1, y_2, ..., y_C]$ is the label of $\boldsymbol{x}$ in a simplex, i.e., $\sum_i y_i = 1$. In the context of adversarial perturbation, we use $\mathcal{S}_\epsilon^{(p)}(\boldsymbol{x}) \stackrel{\text{def}}{=} \{\boldsymbol{\delta} | \|\boldsymbol{\delta}\|_p \leq \epsilon, 0 \leq \boldsymbol{x} + \boldsymbol{\delta} \leq 1\}$ to represent the adversarial budget, i.e., the set of all allowable input perturbations for the input $\boldsymbol{x}$. The adversarial loss function is $\mathcal{L}_\epsilon^{(p)}(\boldsymbol{x}, \boldsymbol{\theta}) \stackrel{\text{def}}{=} \max_{\boldsymbol{\delta} \in \mathcal{S}_\epsilon^{(p)}(\boldsymbol{x})} \mathcal{L}(\boldsymbol{x} + \boldsymbol{\delta}, \boldsymbol{\theta})$. Despite no guarantee to obtain the optimal perturbation in practice, to simplify the notation, we denote the term $\mathcal{L}_\epsilon^{(p)}$ also as the adversarial loss induced by the actual attack algorithms and omit the superscript $(p)$ when there is no ambiguity.

## 2 RELATED WORKS

**Adversarial Attacks:** The existence of adversarial examples is first identified in Szegedy et al. (2013), which focuses on $l_2$ norm-bounded adversarial perturbations. Fast gradient sign method (FGSM) (Goodfellow et al., 2014) introduces an efficient approach by generating perturbations bounded by its $l_\infty$ norm in a single step. Furthermore, projected gradient descent (PGD) (Madry et al., 2018) extends and improves FGSM (Kurakin et al., 2017) by iterative updating and random initialization. In addition to these white-box attacks where the attackers have full access to the models, there are also several black-box attacks (Dong et al., 2018; Andriushchenko et al., 2020) where the attackers' access is restricted. AutoAttack (AA) (Croce & Hein, 2020b) is an ensemble of both white-box and black-box attacks to ensure a more reliable evaluation of model's robustness.

**Adversarial Training:** Adversarial training (Madry et al., 2018; Croce & Hein, 2020a; Sehwag et al.; Rebuffi et al., 2021; Gowal et al., 2021; Rade & Moosavi-Dezfooli, 2021; Cui et al., 2023; Wang et al., 2023) has emerged as a popular and reliable framework to obtain robust models (Athalye et al., 2018; Croce & Hein, 2020b). Under this framework, we first generate adversarial examples and update model parameters based on these examples in each mini-batch update. Different adversarial training variants, such as TRADES (Zhang et al., 2019b) and MART (Wang et al., 2020), may have different loss objective functions for generating adversarial examples and updating model parameters. Furthermore, compared with training on clean inputs, adversarial training is shown to suffer more from overfitting (Rice et al., 2020; Liu et al., 2021a). In this regard, self-adaptive training (SAT) (Huang et al., 2020), which utilizes historical predictions as the soft label, has demonstrated its efficacy in improving the generalization.

**Sparse Perturbations:** Adversarial budget defined by $l_1$ norm is the tightest convex hull of the one defined by $l_0$ norm. In this context, SLIDE (Tramer & Boneh, 2019) extends PGD and employs $k$-coordinate ascent to generate $l_1$ bounded perturbations. Similarly, AutoAttack-$l_1$ (AA-$l_1$) (Croce & Hein, 2021) extends AA to the $l_1$ case. However, AA-$l_1$ is found to generate non-sparse perturbations that SLIDE fails to discover (Jiang et al., 2023), indicating that $l_1$ bounded perturbations are not necessarily sparse. Therefore, we use $l_0$ norm to strictly enforce sparsity. It is challenging to optimize over an adversarial budget defined by $l_0$ norm, because of non-convex adversarial budgets. While naively applying PGD in this case turns out sub-optimal, there are several black-box attacks, including CornerSearch (Croce & Hein, 2019) and Sparse-RS (Croce et al., 2022), and white-box attacks, including Sparse Adversarial and Interpretable Attack Framework (SAIF) (Imtiaz et al., 2022) and Sparse-PGD (sPGD) (Zhong et al., 2024), which address the optimization challenge of finding $l_0$ bounded perturbations. Ultimately, Sparse-AutoAttack (sAA) (Zhong et al., 2024), combining the most potent white-box and black-box attacks, emerges as the most powerful sparse attack.

**Fast Adversarial Training:** While effective, adversarial training is time-consuming due to the use of multi-step attacks. To reduce the computational overhead, some studies (Shafahi et al., 2019; Zhang et al., 2019a) employ faster one-step attacks in adversarial training. However, the training based on these weaker attacks may suffer from catastrophic overfitting (CO) (Kang & Moosavi-Dezfooli, 2021), where the model overfits to these weak attacks instead of achieving true robustness against a variety of attacks. CO is shown to arise from distorted decision boundary caused by *sub-optimal perturbation magnitudes* (Kim et al., 2020). There are several methods proposed to mitigate CO, including aligning the gradients of clean and adversarial samples (Andriushchenko & Flammarion, 2020), adding stronger noise to clean sample (de Jorge Aranda et al., 2022) , adaptive step size (Huang et al., 2023), regularizing abnormal adversarial samples (Lin et al., 2024b), adding layer-wise weight perturbations (Lin et al., 2024a), and penalizing logits discrepancy (Li & Spratling, 2023). Furthermore, compared to its $l_2$ and $l_\infty$ counterparts, CO is caused by overfitting to sparse perturbations during $l_1$ adversarial training (Jiang et al., 2023). To address this issue, Fast-EG-$l_1$ (Jiang et al., 2023) is introduced to generate $l_1$ bounded perturbations by Euclidean geometry instead of coordinate ascent. In this work, we investigate fast adversarial training against $l_0$ bounded perturbations.

## 3 CHALLENGES IN FAST $l_0$ ADVERSARIAL TRAINING

To obtain robust models against sparse perturbations, preliminary efforts use 20-step sPGD in adversarial training, which introduces significant computational overhead. To accelerate training, we

explore using 1-step sPGD in adversarial training. However, as reported in Table 1, the models obtained in this way exhibit weak robustness against stronger and comprehensive sparse attacks such as sAA. In this section, we study the underlying factors that make fast $l_0$ adversarial training challenging by both numerical experiments and theoretical analyses.

### 3.1 Catastrophic Overfitting in Fast $l_0$ Adversarial Training

We plot the learning curves of adversarial training using 1-step sPGD in Figure 1. Specifically, we adopt the multi-$\epsilon$ strategy (Jiang et al., 2023; Zhong et al., 2024) and allow for different adversarial budget sizes, i.e., $\epsilon$, during training and testing. The results in Figure 1 indicate that CO happens in all configurations. Moreover, our observations of CO in $l_0$ cases are different from other cases in several aspects. First, random initialization of adversarial perturbation, proven effective in $l_\infty$, $l_2$ and $l_1$ cases, does not yield similar results in the $l_0$ case. In addition, Figure 1 showcases that the training accuracy on the inputs perturbed by 1-step sPGD is even higher than their clean counterparts. What's more, when CO happens in $l_\infty$, $l_2$ and $l_1$ cases, the model sharply achieves perfect robustness against 1-step attacks but zero robustness against multi-step attacks, both in few mini-batch updates. Such phenomenon is not observed in $l_0$ cases. By contrast, we observe dramatic performance fluctuations on clean examples throughout the training process, even in the fine-tuning phase. Such training instability indicates a non-smooth landscape of the loss function in the parameter space: a subtle change in parameters $\boldsymbol{\theta}$ leads to abrupt fluctuation in the loss.

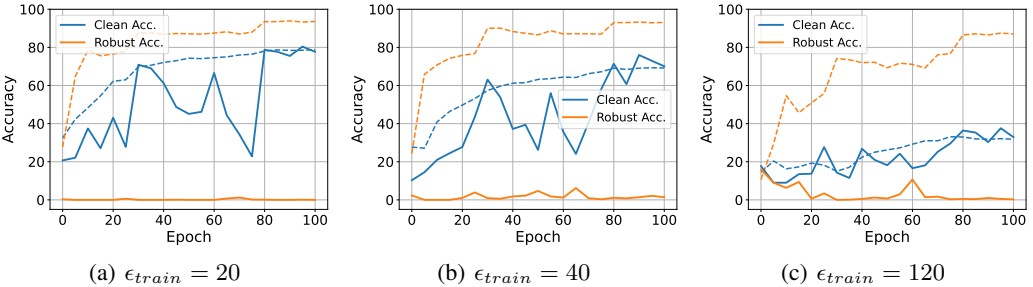

(a) $\epsilon_{train} = 20$        (b) $\epsilon_{train} = 40$        (c) $\epsilon_{train} = 120$

Figure 1: The learning curves of adversarial training against 1-step sPGD (Zhong et al., 2024) with random noise initialization. The models are PreactResNet-18 (He et al., 2016a) trained on CIFAR-10 (Krizhevsky et al., 2009). The dashed and the solid lines represent the accuracy of the training and the test set, respectively. The test robust accuracy is based on sAA with $\epsilon = 20$. The values of $\epsilon$ used in training are shown as $\epsilon_{train}$ in captions, the training robust accuracy is based on the 1-step sPGD with $\epsilon_{train}$.

Table 2: Robust accuracy of the models obtained by 1-step sAT with different $\epsilon_{train}$ against the interpolation between perturbations generated by 1-step sPGD ($\epsilon = 20$) and their corresponding clean examples, where $\alpha$ denotes the interpolation factor, i.e., $\boldsymbol{x}_{interp} = \boldsymbol{x} + \alpha \cdot \boldsymbol{\delta}$. The results of sAA are also reported.

| $\alpha$ | 0.0 | 0.1 | 0.2 | 0.3 | 0.4 | 0.6 | 0.8 | 1.0 | sAA |
|---|---|---|---|---|---|---|---|---|---|
| $\epsilon_{train} = 20$ | 77.5 | 69.8 | **69.1** | 73.7 | 80.4 | 88.0 | 90.2 | 90.4 | **0.0** |
| $\epsilon_{train} = 40$ | 70.2 | **63.1** | 64.3 | 70.9 | 79.8 | 87.4 | 89.6 | 89.6 | **0.0** |
| $\epsilon_{train} = 120$ | 32.5 | 26.5 | **24.5** | 29.4 | 41.5 | 65.2 | 72.8 | 67.6 | **0.0** |

In $l_\infty$ and $l_2$ cases, CO occurs due to distorted decision boundary caused by sub-optimal perturbation magnitudes (Kim et al., 2020). To ascertain if this applies to $l_0$ adversarial training, we evaluate the robustness accuracy of models trained by 1-step sAT with varying $\epsilon_{train}$ against interpolations between the clean inputs and the perturbed ones by 1-step sPGD. Table 2 shows that we cannot find successful adversarial examples through such simple interpolations. In addition, the substantial $l_0$ distance between 1-step sPGD and sAA perturbations (see in Appendix E.1) suggests that CO in $l_0$ adversarial training is primarily due to sub-optimal perturbation locations rather than magnitudes. Consequently, existing CO mitigation methods like GradAlign (Andriushchenko & Flammarion, 2020), ATTA (Zheng et al., 2019), and adaptive step size (Huang et al., 2023) turn out ineffective or insufficient for $l_0$ scenarios. We defer the detailed evaluation to Appendix E.3.

In contrast to other adversarial budgets, $l_0$ adversarial budgets are non-convex, which limits the availability of tools to enhance the quality of the generated perturbations. To address this challenge,

we investigate the loss landscape in the subsequent sections. We show that a smoother loss function can mitigate the negative impact of sub-optimal adversarial perturbations in adversarial training.

## 3.2 Theoretical Analyses on the Smoothness of Adversarial Loss Functions

We first provide theoretical analyses on the smoothness of adversarial loss function. Similar to Liu et al. (2020), we assume the first-order smoothness of the model's outputs $\{f_i\}_{i=0}^{K-1}$.

**Assumption 3.1. (First-order Lipschitz condition)** $\forall i \in \{0, 1, ..., K-1\}$, the function $f_i$ satisfies the following first-order Lipschitz conditions, with constants $L_{\boldsymbol{\theta}}, L_{\boldsymbol{x}}$:

$$\forall \boldsymbol{x}, \boldsymbol{\theta}_1, \boldsymbol{\theta}_2, \quad \|f_i(\boldsymbol{x}, \boldsymbol{\theta}_1) - f_i(\boldsymbol{x}, \boldsymbol{\theta}_2)\| \leq L_{\boldsymbol{\theta}}\|\boldsymbol{\theta}_1 - \boldsymbol{\theta}_2\|, \tag{2}$$

$$\forall \boldsymbol{\theta}, \boldsymbol{x}_1, \boldsymbol{x}_2, \quad \|f_i(\boldsymbol{x}_1, \boldsymbol{\theta}) - f_i(\boldsymbol{x}_2, \boldsymbol{\theta})\| \leq L_{\boldsymbol{x}}\|\boldsymbol{x}_1 - \boldsymbol{x}_2\|. \tag{3}$$

We then study the first-order smoothness of the adversarial loss objective function $\mathcal{L}_\epsilon(\boldsymbol{x}, \boldsymbol{\theta})$.

**Lemma 3.2.** *(Lipschitz continuity of adversarial loss) If Assumption 3.1 holds, we have:*

$$\forall \boldsymbol{x}, \boldsymbol{\theta}_1, \boldsymbol{\theta}_2, \quad \|\mathcal{L}_\epsilon(\boldsymbol{x}, \boldsymbol{\theta}_1) - \mathcal{L}_\epsilon(\boldsymbol{x}, \boldsymbol{\theta}_2)\| \leq A_{\boldsymbol{\theta}}\|\boldsymbol{\theta}_1 - \boldsymbol{\theta}_2\|, \tag{4}$$

*The Lipschitz constant $A_{\boldsymbol{\theta}} = 2\sum_{i \in \mathcal{S}_+} y_i L_{\boldsymbol{\theta}}$ where $\mathcal{S}_+ = \{i \mid y_i > 0, h_i(\boldsymbol{x} + \boldsymbol{\delta}_1, \boldsymbol{\theta}_2) > h_i(\boldsymbol{x} + \boldsymbol{\delta}_1, \boldsymbol{\theta}_1)\}$, $\boldsymbol{\delta}_1 \in \arg\max_{\boldsymbol{\delta} \in \mathcal{S}_\epsilon} \mathcal{L}(\boldsymbol{x} + \boldsymbol{\delta}, \boldsymbol{\theta})$ and $\boldsymbol{\delta}_2 \in \arg\max_{\boldsymbol{\delta} \in \mathcal{S}_\epsilon} \mathcal{L}(\boldsymbol{x} + \boldsymbol{\delta}, \boldsymbol{\theta})$.*

The proof is deferred to Appendix B.1, in which we can see the upper bound in Lemma 3.2 is tight and can be achieved in the worst cases. Lemma 3.2 indicates that the adversarial loss $\mathcal{L}_\epsilon(\boldsymbol{x}, \boldsymbol{\theta})$ is Lipschitz continuous, which is consistent with Liu et al. (2020).

To study the second-order smoothness of $\mathcal{L}_\epsilon(\boldsymbol{x}, \boldsymbol{\theta})$, we start with the following assumption.

**Assumption 3.3. (Second-order Lipschitz condition)** $\forall i \in \{0, 1, ..., K-1\}$, the function $f_i$ satisfies the following second-order Lipschitz conditions, with constants $L_{\boldsymbol{\theta}\boldsymbol{\theta}}, L_{\boldsymbol{\theta}\boldsymbol{x}}$:

$$\forall \boldsymbol{x}, \boldsymbol{\theta}_1, \boldsymbol{\theta}_2, \quad \|\nabla_{\boldsymbol{\theta}} f_i(\boldsymbol{x}, \boldsymbol{\theta}_1) - \nabla_{\boldsymbol{\theta}} f_i(\boldsymbol{x}, \boldsymbol{\theta}_2)\| \leq L_{\boldsymbol{\theta}\boldsymbol{\theta}}\|\boldsymbol{\theta}_1 - \boldsymbol{\theta}_2\|, \tag{5}$$

$$\forall \boldsymbol{\theta}, \boldsymbol{x}_1, \boldsymbol{x}_2, \quad \|\nabla_{\boldsymbol{\theta}} f_i(\boldsymbol{x}_1, \boldsymbol{\theta}) - \nabla_{\boldsymbol{\theta}} f_i(\boldsymbol{x}_2, \boldsymbol{\theta})\| \leq L_{\boldsymbol{\theta}\boldsymbol{x}}\|\boldsymbol{x}_1 - \boldsymbol{x}_2\|. \tag{6}$$

**Lemma 3.4.** *(Lipschitz smoothness of adversarial loss) If Assumption 3.1 and 3.3 hold, we have:*

$$\forall \boldsymbol{x}, \boldsymbol{\theta}_1, \boldsymbol{\theta}_2, \quad \|\nabla_{\boldsymbol{\theta}}\mathcal{L}_\epsilon(\boldsymbol{x}, \boldsymbol{\theta}_1) - \nabla_{\boldsymbol{\theta}}\mathcal{L}_\epsilon(\boldsymbol{x}, \boldsymbol{\theta}_2)\| \leq A_{\boldsymbol{\theta}\boldsymbol{\theta}}\|\boldsymbol{\theta}_1 - \boldsymbol{\theta}_2\| + B_{\boldsymbol{\theta}\boldsymbol{\delta}}. \tag{7}$$

*The Lipschitz constant $A_{\boldsymbol{\theta}\boldsymbol{\theta}} = L_{\boldsymbol{\theta}\boldsymbol{\theta}}$ and $B_{\boldsymbol{\theta}\boldsymbol{\delta}} = L_{\boldsymbol{\theta}\boldsymbol{x}}\|\boldsymbol{\delta}_1 - \boldsymbol{\delta}_2\| + 4L_{\boldsymbol{\theta}}$ where $\boldsymbol{\delta}_1 \in \arg\max_{\boldsymbol{\delta} \in \mathcal{S}_\epsilon} \mathcal{L}(\boldsymbol{x} + \boldsymbol{\delta}, \boldsymbol{\theta}_1)$ and $\boldsymbol{\delta}_2 \in \arg\max_{\boldsymbol{\delta} \in \mathcal{S}_\epsilon} \mathcal{L}(\boldsymbol{x} + \boldsymbol{\delta}, \boldsymbol{\theta}_2)$.*

The proof is deferred to Appendix B.2. Lemma 3.4 indicates the adversarial loss objective function $\mathcal{L}_\epsilon(\boldsymbol{x}, \boldsymbol{\theta})$ w.r.t. the model parameter $\boldsymbol{\theta}$ is no longer smooth. That is to say, gradients in arbitrarily small neighborhoods in the $\boldsymbol{\theta}$-space can change discontinuously. Furthermore, the degree of discontinuity is indicated by the value of $B_{\boldsymbol{\theta}\boldsymbol{\delta}}$. Given the expression of $B_{\boldsymbol{\theta}\boldsymbol{\delta}}$, we can conclude that a larger $\|\boldsymbol{\delta}_1 - \boldsymbol{\delta}_2\|$ can intensify the gradient discontinuity. Additionally, as elucidated by *Theorem 2* in Liu et al. (2020), the gradients are non-vanishing in adversarial training. A large $B_{\boldsymbol{\theta}\boldsymbol{\delta}}$ introduces large gradient magnitudes asymptotically, making optimization challenging.

However, in practice, we may use non-smooth activations, like ReLU, which do not strictly satisfy Assumption 3.3. For example, the gradient of ReLU changes abruptly in the neighborhood around 0. In this regard, we provide a more detailed analysis of this case in Appendix C, which suggests that our analyses can be straightforwardly extended to networks with non-smooth activations.

Without the loss of generality, the Lipschitz properties in Assumption 3.1 and 3.3 can be based on any *proper* $l_p$ norm, i.e., $p \in [1, +\infty]$, which, however, does not include $l_0$ norm. Correspondingly, $\|\boldsymbol{\delta}_1 - \boldsymbol{\delta}_2\|$ in the expression of $B_{\boldsymbol{\theta}\boldsymbol{\delta}}$ is based on the same norm as in the assumptions. On the popular benchmark CIFAR-10, the commonly used values of $\epsilon$ in the $l_0$, $l_1$, $l_2$ and $l_\infty$ cases are $360^1$, 24, 0.5 and $8/255$, respectively (Madry et al., 2018; Croce & Hein, 2021; Jiang et al., 2023; Zhong et al., 2024). In Appendix D, we discuss the numerical upper bound of $\|\boldsymbol{\delta}_1 - \boldsymbol{\delta}_2\|$ when the Lipschitz assumptions are based on different proper norms. The results demonstrate that the upper bound of $\|\boldsymbol{\delta}_1 - \boldsymbol{\delta}_2\|$ in the $l_0$ case is always significantly larger than other cases, indicating a more craggy adversarial loss function in $l_0$ adversarial training. Moreover, to corroborate the Lipschitz smoothness assumption in Inequality (6), we compare the distances between the gradients induced by one-step and multi-step attacks with different adversarial budgets in Appendix E.2.

---

[1]In Zhong et al. (2024), the $l_0$ adversarial budget for training on CIFAR-10 is 120 *in the pixel space* of RGB images, so the $l_0$ norm *in the feature space* is 360.

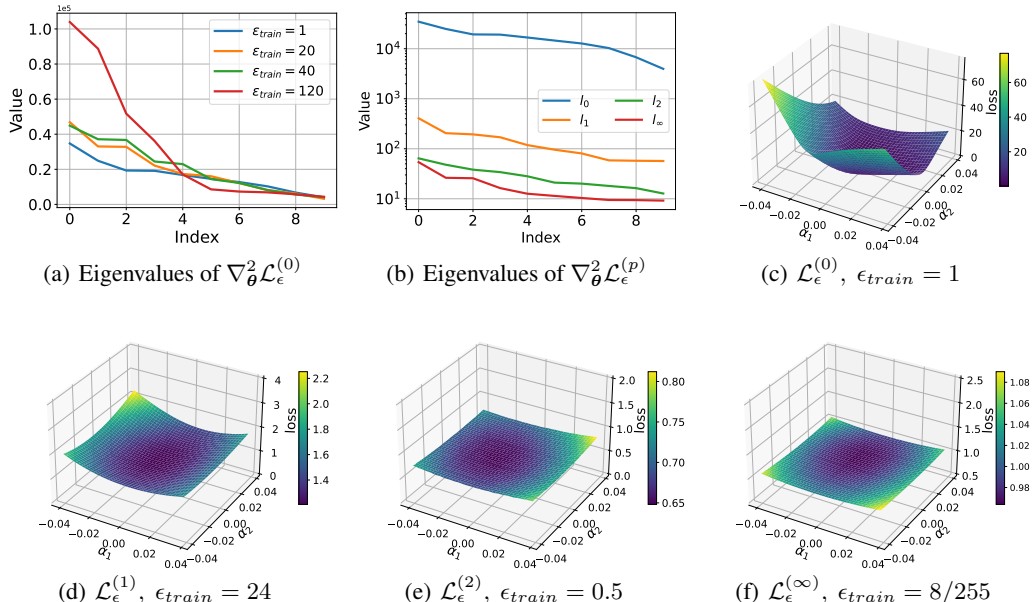

(a) Eigenvalues of $\nabla^2_{\boldsymbol{\theta}} \mathcal{L}^{(0)}_{\epsilon}$      (b) Eigenvalues of $\nabla^2_{\boldsymbol{\theta}} \mathcal{L}^{(p)}_{\epsilon}$      (c) $\mathcal{L}^{(0)}_{\epsilon}$, $\epsilon_{train} = 1$

(d) $\mathcal{L}^{(1)}_{\epsilon}$, $\epsilon_{train} = 24$      (e) $\mathcal{L}^{(2)}_{\epsilon}$, $\epsilon_{train} = 0.5$      (f) $\mathcal{L}^{(\infty)}_{\epsilon}$, $\epsilon_{train} = 8/255$

Figure 2: Smoothness of adversarial loss objective functions under different settings. All losses are calculated on the training set of CIFAR-10 (Krizhevsky et al., 2009) by PreactResNet-18 (He et al., 2016a). The $l_0$, $l_1$, $l_2$ and $l_\infty$ models are obtained by 1-step sAT (Zhong et al., 2024), Fast-EG-$l_1$ (Jiang et al., 2023), 1-step PGD (Rice et al., 2020) and GradAlign (Andriushchenko et al., 2020), respectively. **(a)** Top 10 eigenvalues of $\nabla^2_{\boldsymbol{\theta}} \mathcal{L}^{(0)}_{\epsilon}(\boldsymbol{x}, \boldsymbol{\theta})$ with different values of $\epsilon_{train}$ in the $l_0$ case. **(b)** Top 10 eigenvalues of $\nabla^2_{\boldsymbol{\theta}} \mathcal{L}^{(p)}_{\epsilon}(\boldsymbol{x}, \boldsymbol{\theta})$ under different choices of $p$, including $l_0$ ($\epsilon_{train} = 1$), $l_1$ ($\epsilon_{train} = 24$), $l_2$ ($\epsilon_{train} = 0.5$) and $l_\infty$ ($\epsilon_{train} = 8/255$). The y-axis is shown in the log scale. **(c) - (f)** The loss landscape of $\mathcal{L}_{\epsilon}(\boldsymbol{x}, \boldsymbol{\theta} + \alpha_1 \boldsymbol{v}_1 + \alpha_2 \boldsymbol{v}_2)$ where $\boldsymbol{v}_1$ and $\boldsymbol{v}_2$ are the eigenvectors associated with the top 2 eigenvalues of $\nabla^2_{\boldsymbol{\theta}} \mathcal{L}_{\epsilon}(\boldsymbol{x}, \boldsymbol{\theta})$, respectively. The y-scales for different sub-figures are different. **(c)** $l_0$ case, $\epsilon_{train} = 1$. **(d)** $l_1$ case, $\epsilon_{train} = 24$. **(e)** $l_2$ case, $\epsilon_{train} = 0.5$. **(f)** $l_\infty$ case, $\epsilon_{train} = 8/255$.

### 3.3 NUMERICAL ANALYZES ON THE SMOOTHNESS OF ADVERSARIAL LOSS FUNCTIONS

To validate the conclusions in theoretical analyses, we conduct numerical experiments to study the properties of loss landscape of $l_0$ adversarial training and compare it with the $l_\infty$, $l_2$ and $l_1$ cases.

We first study the curvature in the neighborhood of model parameters, which reflects the second-order smoothness of the loss function and is dominated by top eigenvalues of Hessian matrix $\nabla^2_{\boldsymbol{\theta}} \mathcal{L}_{\epsilon}(\boldsymbol{x}, \boldsymbol{\theta})$. Numerically, we employ the power method (Yao et al., 2018; Liu et al., 2020; Zhong & Liu, 2023) to iteratively estimate the eigenvalues and the corresponding eigenvectors of Hessian matrices. We plot the top-10 eigenvalues of the Hessian matrices $\nabla^2_{\boldsymbol{\theta}} \mathcal{L}_{\epsilon}(\boldsymbol{x}, \boldsymbol{\theta})$ under different $\epsilon$ in $l_0$ cases in Figure 2 (a). In addition, we compare the Hessian spectrum in the $l_0$ case with $l_\infty$, $l_2$ and $l_1$ cases in Figure 2 (b). Our results in Figure 2 (a) demonstrate that eigenvalues of Hessian matrices in $l_0$ cases increase as $\epsilon$ grows, indicating a higher degree of non-smoothness for a larger $\epsilon$. Moreover, Figure 2 (b) indicates that the adversarial loss landscape in the $l_0$ case is more craggy than its $l_\infty$, $l_2$ and $l_1$ counterparts, even when we set $\epsilon = 1$, i.e., perturbing only a single pixel. These observations corroborate that $l_0$ adversarial training exhibits worse second-order smoothness than other cases.

To study the first-order smoothness, we visualize the loss landscape of different settings in Figures 2 (c)-(f), which demonstrate that the loss in the $l_0$ case abruptly increases even with subtle changes in the model parameters. This further suggests the non-smooth nature of the $l_0$ adversarial loss landscape. More loss landscape visualizations of $l_0$ adversarial training with different $\epsilon$ are provided in Appendix E.7. The observations are consistent with that in Figure 2. Accordingly, we confirm that the loss landscape of $l_0$ adversarial loss function is more craggy than other cases from both theoretical and empirical perspectives. In addition, among the cases studied in Figure 3, the $l_0$ cases are the only ones suffering from CO, while the $l_\infty$, $l_2$ and $l_1$ cases do not. This indicates that the craggy loss landscape aggravates CO.

On the other side, we show in Figure 3 that successful attempts to obtain robust models against $l_0$ bounded perturbation also include elements that help improve the smoothness of the loss landscape.

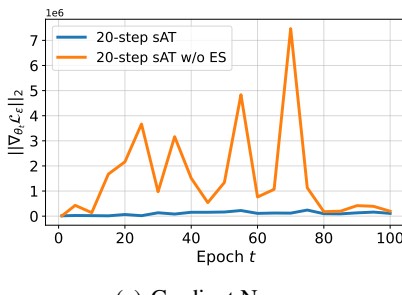 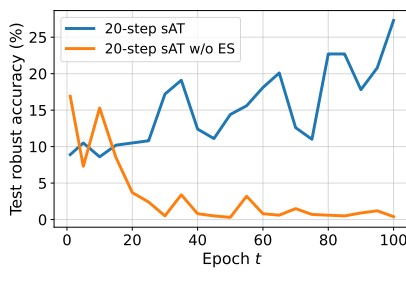

(a) Gradient Norm          (b) Test Robust Accuracy

Figure 3: Relationship between craggy loss landscape and CO. **(a)** Gradient norm $\|\nabla_{\boldsymbol{\theta}_t}\mathcal{L}_\epsilon\|_2$, which indicates the first-order smoothness of $\mathcal{L}_\epsilon$. **(b)** Test robust accuracy against sAA ($\epsilon = 20$). The results are obtained from PreactResNet-18 trained on CIFAR-10, where $\epsilon_{train} = 40$. Note that since the training of 20-step sAT w/o ES diverges under $\epsilon_{train} = 120$, the results are presented under $\epsilon_{train} = 40$ instead.

20-step sAT in Zhong et al. (2024) uses an early stopping (ES) strategy to avoid CO and to achieve competitive performance. Specifically, ES interrupts the attack iteration once the current perturbed input is misclassified. ES is shown to circumvent the potential for excessive gradient magnitude while maintaining the efficacy of the generated perturbations. Figure 3 compares the cases with and without ES in terms of gradient norm and robust accuracy on the test set by sAA. We can observe from Figure 3 that 20-step sAT without ES still suffer from CO and the corresponding gradient magnitude during training indicates a craggy loss landscape. This finding further highlights a strong correlation between CO and the craggy nature of the loss landscape in $l_0$ adversarial training.

In summary, our results suggest that the $l_0$ adversarial training exhibits a more craggy loss landscape than other cases, which shows a strong correlation with CO. Additionally, despite the non-trivial performance of 20-step sAT with ES, its performance still exhibits considerable fluctuation and can be further improved, underscoring the need for a smoother loss function. In the next section, we will propose our method to address the CO issue in fast $l_0$ adversarial training.

## 4 SOFT LABEL AND TRADE-OFF LOSS SMOOTH ADVERSARIAL LOSS

Notice that $A_{\boldsymbol{\theta}}$ in Lemma 3.2 can be regarded as a function of the label $\boldsymbol{y}$. Thus, we first study how different $\boldsymbol{y}$ affects the properties of the adversarial loss objective function $\mathcal{L}_\epsilon(\boldsymbol{x}, \boldsymbol{\theta})$. Let $\boldsymbol{y}_h \in \{0,1\}^K$ and $\boldsymbol{y}_s \in (0,1)^K$ denote the hard and soft label, respectively. That is to say, $\boldsymbol{y}_h$ is a one-hot vector, while $\boldsymbol{y}_s$ is a dense vector in a simplex. Then, we have the following theorem:

**Theorem 4.1.** *(Soft label improves Lipschitz continuity) Based on Lemma 3.2, given a hard label vector $\boldsymbol{y}_h \in \{0,1\}^K$ and a soft label vector $\boldsymbol{y}_s \in (0,1)^K$, we have $A_{\boldsymbol{\theta}}(\boldsymbol{y}_s) \leq A_{\boldsymbol{\theta}}(\boldsymbol{y}_h)$.*

The proof is deferred to Appendix B.3. Theorem 4.1 indicates that soft labels lead to a reduced first-order Lipschitz constant, thereby enhancing the Lipschitz continuity of the adversarial loss function. However, as indicated by Lemma 3.4, the second-order Lipschitz constant remains unaffected by variations in $\boldsymbol{y}$. Considering the poor performance on clean inputs when CO happens, we introduce a trade-off loss objective function $\mathcal{L}_{\epsilon,\alpha}$ which interpolates between the loss on the clean inputs and that on the adversarial inputs.

$$\mathcal{L}_{\epsilon,\alpha}(\boldsymbol{x}, \boldsymbol{\theta}) = (1-\alpha)\mathcal{L}(\boldsymbol{x}, \boldsymbol{\theta}) + \alpha \max_{\boldsymbol{\delta} \in \mathcal{S}_\epsilon(\boldsymbol{x})} \mathcal{L}(\boldsymbol{x}+\boldsymbol{\delta}, \boldsymbol{\theta}) \tag{8}$$

where $\alpha \in [0,1]$ is the interpolation factor. Then, we have the following theorem:

**Theorem 4.2.** *(Trade-off loss function improves Lipschitz smoothness) If Assumption 3.1 and 3.3 hold, we have:*

$$\|\nabla_{\boldsymbol{\theta}}\mathcal{L}_{\epsilon,\alpha}(\boldsymbol{x}, \boldsymbol{\theta}_1) - \nabla_{\boldsymbol{\theta}}\mathcal{L}_{\epsilon,\alpha}(\boldsymbol{x}, \boldsymbol{\theta}_2)\| \leq A_{\boldsymbol{\theta}\boldsymbol{\theta}}\|\boldsymbol{\theta}_1 - \boldsymbol{\theta}_2\| + B'_{\boldsymbol{\theta}\boldsymbol{\delta}} \tag{9}$$

*The Lipschitz constant $A_{\boldsymbol{\theta}\boldsymbol{\theta}} = L_{\boldsymbol{\theta}\boldsymbol{\theta}}$ and $B'_{\boldsymbol{\theta}\boldsymbol{\delta}} = \alpha L_{\boldsymbol{\theta}\boldsymbol{x}}\|\boldsymbol{\delta}_1 - \boldsymbol{\delta}_2\| + 2(1+\alpha)L_{\boldsymbol{\theta}}$ where $\boldsymbol{\delta}_1 \in \arg\max_{\boldsymbol{\delta} \in \mathcal{S}_\epsilon(\boldsymbol{x})}\mathcal{L}(\boldsymbol{x}+\boldsymbol{\delta}, \boldsymbol{\theta}_1)$ and $\boldsymbol{\delta}_2 \in \arg\max_{\boldsymbol{\delta} \in \mathcal{S}_\epsilon(\boldsymbol{x})}\mathcal{L}(\boldsymbol{x}+\boldsymbol{\delta}, \boldsymbol{\theta}_2)$.*

The proof is deferred to Appendix B.4. According to Theorem 4.2, the trade-off loss function $\mathcal{L}_{\epsilon,\alpha}$ enhances the second-order smoothness of adversarial loss objective function. The interpolation

factor $\alpha$ controls the balance between the loss on the clean inputs and the loss on the adversarial inputs. On one hand, a smaller value of $\alpha$ results in a smoother loss objective function, but it assigns less weight to the loss of the adversarial inputs and potentially hurts the robustness of the obtained model. On the other hand, a bigger value of $\alpha$ assigns more weight to the adversarial loss to focus on robustness, but it makes the corresponding adversarial loss objective function more challenging for optimization. Furthermore, compared with $l_1$, $l_2$ and $l_\infty$ cases, the trade-off loss function is particularly useful and necessary in the $l_0$ case. This is supported by the analyses in Section 3.2 and Appendix D, which demonstrate that $\|\boldsymbol{\delta}_1 - \boldsymbol{\delta}_2\|$ is much larger in $l_0$ bounded perturbations than other cases. Therefore, we expect the trade-off loss function $\mathcal{L}_{\epsilon,\alpha}$ can help mitigate CO by improving smoothness.

Similar to Lemma 3.4, Theorem 4.2 can be straightforwardly extended to the networks with non-smooth activations, where Assumption 3.3 is not strictly satisfied. We provide a more detailed analysis in Appendix C to demonstrate the generality of our conclusions.

In summary, soft labels and the trade-off loss function can improve the first-order and second-order smoothness, respectively. Therefore, we can stabilize and improve the performance of fast adversarial training against $l_0$ bounded perturbations by combining both techniques together.

Among various approaches available, we mainly exploit trade-off loss function, self-adaptive training (SAT) (Huang et al., 2020) and TRADES (Zhang et al., 2019b). Specifically, SAT utilizes the moving average of previous predictions as the soft label to calculate the loss. TRADES combines the soft label and the trade-off loss function. It utilizes the trade-off loss function to balance the clean and robust accuracy and employs the prediction on the clean inputs as the soft label when calculating the loss for adversarial inputs. In Appendix A, we provide the pseudo-codes of both SAT and TRADES and the formulation of their combination as a reference.

## 5 EXPERIMENTS

In this section, we perform extensive experiments to investigate various approaches that can stabilize and improve the performance of fast adversarial training against $l_0$ bounded perturbations. Furthermore, we compare the performance of 1-step adversarial training with the multi-step counterpart on different datasets. Our results demonstrate that approaches combining soft labels and trade-off loss function significantly enhance the stability and efficacy of 1-step adversarial training, even surpassing some baselines of multi-step adversarial training. Finally, we validate the efficacy of our method on different networks in Appendix E.6, visualize the loss landscape when using soft label and trade-off loss function in Appendix E.8 to demonstrate its improved smoothness, and conduct ablation studies for analysis in Appendix E.9.

### 5.1 APPROACHES TO IMPROVING 1-STEP $l_0$ ADVERSARIAL TRAINING

Table 3: Comparison of different approaches and their combinations in robust accuracy (%) by sAA. The target sparsity level $\epsilon = 20$. We compare PreAct ResNet-18 (He et al., 2016a) models trained on CIFAR-10 (Krizhevsky et al., 2009) with 100 epochs. The *italic numbers* indicate catastrophic overfitting (CO) happens.

| Method | sAT | Tradeoff | sTRADES (T) | sTRADES (F) |
|---|---|---|---|---|
| 1-step | *0.0* | *2.6* | 31.0 | 55.4 |
| + N-FGSM | *0.3* | *17.5* | 46.9 | 55.9 |
| + SAT | 29.3 | 30.3 | 61.4 | 59.4 |
| + SAT & N-FGSM | **43.8** | **39.2** | **63.0** | **62.6** |

We begin our analysis by evaluating the effectiveness of different approaches and their combinations, focusing on those that incorporate either soft labels or trade-off loss functions. Additionally, we explore the data augmentation technique N-FGSM (de Jorge Aranda et al., 2022), known for its ability to improve the performance of fast adversarial training without imposing significant computational overhead. Our findings, summarized in Table 3, are all based on 1-step adversarial training. The robust accuracy is measured using the sparse-AutoAttack (sAA) method, with $\epsilon$ set to 20.

In Table 3, we investigate the following approaches and their combinations: **(1) sAT**: adversarial training against 1-step sPGD (Zhong et al., 2024). **(2) Tradeoff**: 1-step adversarial training with the trade-off loss function defined in Eq. (8). **(3) sTRADES**: the 1-step sTRADES (Zhong et al.,

2024). As discussed in Appendix A, it incorporates both soft label and trade-off loss function. We include two variants of sTRADES for comparison: **sTRADES (T)** is the training mode where we only use the loss objective function of TRADES for training but still use the cross-entropy loss to generate adversarial examples; **sTRADES (F)** is the full mode where we use the loss objective function of TRADES for both training and generating adversarial perturbations. Compared with 1-step sAT, sTRADES (T) introduces $25\%$ overhead while sTRADES (F) introduces $50\%$ overhead. **(4) SAT**: self-adaptive training (Huang et al., 2020). As discussed in Appendix A, it introduces soft labels based on the moving average of the historical predictions and uses adaptive weights for training instances of different prediction confidence. SAT can be incorporated into sAT, Tradeoff and sTRADES. **(5) N-FGSM**: data augmentation technique by adding random noise to the training data. It is proven effective in 1-step adversarial training (de Jorge Aranda et al., 2022). N-FGSM can be incorporated into sAT, Tradeoff, sTRADES and used jointly with SAT. The implementation details are deferred to Appendix F.

The results in Table 3 indicate that using trade-off loss function alone still suffers from CO. In contrast, using soft label, either by SAT or sTRADES, can eliminate CO and achieve notable robust accuracy. This suggests that the soft label has a more prominent role in mitigating overfitting than the trade-off loss function in 1-step $l_0$ adversarial training. Furthermore, sTRADES (F) alone outperforms sTRADES (T) along by a substantial margin of $24.4\%$, which can be attributed to the generation of higher-quality adversarial examples for training by sTRADES (F). Finally, both SAT and N-FGSM can enhance the performance of all approaches, demonstrating their effectiveness.

It is important to note that all the results presented in Table 3 are obtained using sAA, which is known for generating the strongest attacks in terms of sparse perturbations. Our findings demonstrate that incorporating soft labels and trade-off loss function yields substantial performance improvements in 1-step $l_0$ adversarial training. Among various combinations of methods explored, the model trained with sTRADES (T) in combination with SAT and N-FGSM achieves the highest robust accuracy against sAA, reaching an impressive $63.0\%$. This establishes a new state-of-the-art performance in the context of fast robust learning methods against $l_0$ bounded perturbations. For convenience, we name this combination (i.e., 1-step sTRADES + SAT + N-FGSM) **Fast-L**oss **S**moothing-$l_0$ (**Fast-LS-**$l_0$) in the subsequent sections. Its pseudo-code is given in Algorithm 3 of Appendix A. Additionally, the comparison with more baselines that either mitigate CO or smooth the loss function is undertaken in Appendix E.3. The results demonstrate that our method is the most effective approach for fast $l_0$ adversarial training.

## 5.2 Comparison with Multi-Step Adversarial Training

In this section, we compare 1-step adversarial training with its multi-step counterpart. For multi-step adversarial training, we follow the settings in Zhong et al. (2024) and use 20-step sPGD based on cross-entropy to generate adversarial perturbations in sAT and sTRADES. Similar to Table 3, we incorporate SAT and N-FGSM into multi-step adversarial training as well. For 1-step adversarial training, we focus on the configurations with the best performance in Table 3, i.e., Fast-LS-$l_0$.

We conduct extensive experiments on various datasets. The results on CIFAR-10 (Krizhevsky et al., 2009) and ImageNet-100 (Deng et al., 2009) are demonstrated in Table 4. More results on CIFAR-100 (Krizhevsky et al., 2009) and GTSRB (Stallkamp et al., 2012) are in Table 7 and 8 of Appendix E.4, respectively. Following the settings in (Zhong et al., 2024), and given the prohibitively high complexity involved, we exclude multi-step sTRADES from the evaluation on ImageNet-100. In addition to the performance under sAA, we report the robust accuracy of these models under various black-box and white box attacks, including CornerSearch (CS) (Croce & Hein, 2019), Sparse-RS (RS) (Croce et al., 2022), SAIF (Imtiaz et al., 2022) and two versions of sPGD (Zhong et al., 2024). Note that, we do not include SparseFool (Modas et al., 2019) and PGD_$0$ (Croce & Hein, 2019) for evaluation, because they only have trivial attack success rate on our models. Moreover, we report the clean accuracy and the total running time for reference. Finally, to more comprehensively validate the effectiveness of our results, we run the experiments for multiple times and report the standard deviation of the performance in Table 9 of Appendix E.5.

The results in Table 4, 7 and 8 suggest that both soft labels and trade-off loss function, introduced by either SAT or TRADES, can improve the performance of both 1-step adversarial training and multi-step adversarial training. In addition, N-FGSM, originally designed for one-step adversarial

Table 4: Robust accuracy (%) of various models against various attacks that generate $l_0$ bounded perturbations on different datasets. **(a)** The models are PreAct ResNet-18 trained on **CIFAR-10**, where the sparsity level $\epsilon = 20$. CornerSearch (CS) is evaluated on 1000 samples due to its high computational complexity. **(b)** The models are ResNet-34 trained on **ImageNet-100**, where the sparsity level $\epsilon = 200$. CS is not evaluated here due to its high computational complexity, i.e. nearly 1 week on one GPU for each run. Note that **S** and **N** denote SAT and N-FGSM, respectively. The results of vanilla 20-step sAT and sTRADES are obtained from (Zhong et al., 2024). All experiments are implemented on one NVIDIA RTX 6000 Ada GPU.

(a) **CIFAR-10,** $\epsilon = 20$

| Model | Time Cost | Clean | Black-Box | | SAIF | White-Box | | sAA |
|---|---|---|---|---|---|---|---|---|
| | | | CS | RS | | $sPGD_{proj}$ | $sPGD_{unproj}$ | |
| *Multi-step* | | | | | | | | |
| sAT | 5h 16m | 84.5 | 52.1 | 36.2 | 76.6 | 75.9 | 75.3 | 36.2 |
| +S | 5h 24m | 80.4 | 58.4 | 55.7 | 75.0 | 75.1 | 74.0 | 55.5 |
| +S&N | 5h 28m | 80.8 | 64.1 | 61.1 | 76.1 | 76.8 | 75.1 | 61.0 |
| sTRADES | 5h 30m | 89.8 | 69.9 | 61.8 | 84.9 | 84.6 | 81.7 | 61.7 |
| +S | 5h 27m | 86.7 | 71.1 | 65.1 | 82.2 | 79.9 | 77.8 | 64.1 |
| +S&N | 5h 22m | 82.2 | 66.3 | 66.1 | 77.1 | 74.1 | 72.2 | **65.5** |
| *One-step* | | | | | | | | |
| **Fast-LS-$l_0$ (T)** | 50m | 82.5 | 69.3 | 65.4 | 75.7 | 67.2 | 67.7 | **63.0** |
| **Fast-LS-$l_0$ (F)** | 59m | 82.6 | 69.6 | 64.1 | 75.2 | 64.6 | 68.4 | 62.6 |

(b) **ImageNet,** $\epsilon = 200$

| Model | Time Cost | Clean | Black-Box | | SAIF | White-Box | | sAA |
|---|---|---|---|---|---|---|---|---|
| | | | CS | RS | | $sPGD_{proj}$ | $sPGD_{unproj}$ | |
| *Multi-step* | | | | | | | | |
| sAT | 324h 57m | 86.2 | - | 61.4 | 69.0 | 78.0 | 77.8 | 61.2 |
| +S | 337h 07m | 83.2 | - | 71.8 | 75.0 | 78.8 | 77.2 | 71.4 |
| +S&N | 336h 20m | 83.0 | - | 75.0 | 76.4 | 78.8 | 79.2 | 74.8 |
| sTRADES | 358h 55m | 84.8 | - | 76.0 | 77.4 | 80.6 | 81.4 | 75.8 |
| +S | 359h 39m | 82.8 | - | 78.2 | 79.2 | 80.6 | 80.0 | **78.2** |
| +S&N | 359h 55m | 82.4 | - | 78.2 | 79.2 | 78.2 | 79.8 | 77.8 |
| *One-step* | | | | | | | | |
| **Fast-LS-$l_0$ (T)** | 43h 48m | 82.4 | - | 76.8 | 75.4 | 74.6 | 74.6 | 72.4 |
| **Fast-LS-$l_0$ (F)** | 55h 39m | 80.0 | - | 77.4 | 76.0 | 76.6 | 74.4 | **72.8** |

training, also contributes to performance improvements in the multi-step scenario. Furthermore, these techniques can greatly narrow down the performance gaps between 1-step and multi-step adversarial training, making fast adversarial training more feasible and competitive in the context of sparse perturbations. With the assistance of SAT and N-FGSM, our Fast-LS-$l_0$ can achieve a performance that is merely 2.5% lower than that of the 20-step sTRADES while requiring less than 1/6 of the total running time.

# 6 CONCLUSION

In this paper, we highlight the catastrophic overfitting (CO) in the fast $l_0$ adversarial training is induced by sub-optimal perturbation locations of 1-step attacks, which is distinct from the $l_\infty$, $l_2$ and $l_1$ cases. Theoretical and empirical analyses reveal that the loss landscape of $l_0$ adversarial training is more craggy than other cases, and the craggy loss landscape strongly correlates with CO. To address these issues, we propose Fast-LS-$l_0$ that incorporates soft label and trade-off loss function to smooth the adversarial loss function. Extensive experiments demonstrate the effectiveness of our method in mitigating CO and narrowing down the performance gap between 1-step and multi-step $l_0$ adversarial training. The models trained with our method exhibit state-of-the-art robustness against sparse attacks in the context of fast adversarial training.

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

# A  ALGORITHM DETAILS

---

**Algorithm 1** Self-Adaptive Training (SAT) (Huang et al., 2020)

---

1: **Input:** Data: $\{(\boldsymbol{x}_i, \boldsymbol{y}_i)\}_n$; Initial target $\{\boldsymbol{t}_i\}_n = \{\boldsymbol{y}_i\}_n$; Batch size: $m$; Classifier: $f$; Enabling epoch: $E_s$; Momentum factor: $\alpha$
2: **repeat**
3:  Fetch mini-batch data $\{(\boldsymbol{x}_i, \boldsymbol{t}_i)\}_m$ at current epoch $e$
4:  **for** $i = 1, ..., m$ **do**
5:    $\boldsymbol{p}_i = \text{softmax}(f(\boldsymbol{x}_i))$
6:    **if** $e > E_s$ **then**
7:      $\boldsymbol{t}_i = \alpha \times \boldsymbol{t}_i + (1 - \alpha) \times \boldsymbol{p}_i$
8:    **end if**
9:    $w_i = \max_j \boldsymbol{t}_{i,j}$
10:  **end for**
11:  Calculate the loss $\mathcal{L}_{SAT} = -\frac{1}{\sum_i w_i} \sum_i w_i \sum_j \boldsymbol{t}_{i,j} \log \boldsymbol{p}_{i,j}$
12:  Update the parameters of $f$ on $\mathcal{L}_{SAT}$
13: **until** end of training

---

---

**Algorithm 2** TRADES (Zhang et al., 2019b)

---

1: **Input:** Data: $(\boldsymbol{x}, \boldsymbol{y})$; Classifier: $f$; Balancing factor: $\beta$; TRADES mode: $mode$; Sparse level: $\epsilon$
2: **if** $mode = F$ **then**
3:  Generate adversarial sample $\widetilde{\boldsymbol{x}} = \max_{(\widetilde{\boldsymbol{x}} - \boldsymbol{x}) \in \mathcal{S}_\epsilon(\boldsymbol{x})} \text{KL}(f(\boldsymbol{x}), f(\widetilde{\boldsymbol{x}}))$
4: **else if** $mode = T$ **then**
5:  Generate adversarial sample $\widetilde{\boldsymbol{x}} = \max_{(\widetilde{\boldsymbol{x}} - \boldsymbol{x}) \in \mathcal{S}_\epsilon(\boldsymbol{x})} \text{CE}(f(\widetilde{\boldsymbol{x}}), \boldsymbol{y})$
6: **end if**
7: Calculate the loss $\mathcal{L}_{TRADES} = \text{CE}(f(\boldsymbol{x}), \boldsymbol{y}) + \beta \cdot \text{KL}(f(\boldsymbol{x}), f(\widetilde{\boldsymbol{x}}))$
8: Update the parameters of $f$ on $\mathcal{L}_{TRADES}$

---

The pseudo-codes of SAT (Huang et al., 2020) and TRADES (Zhang et al., 2019b) are provided in Algorithm 1 and 2, respectively. For SAT, the moving average of the previous predictions $\{\boldsymbol{t}_i\}^n$ can be regarded as the soft labels. For TRADES, $f(\boldsymbol{x})$ can be seen as the soft label of $f(\widetilde{\boldsymbol{x}})$, and the combination of cross-entropy and KL divergence is also a trade-off loss function. Note that when combining SAT and TRADES, the loss $\mathcal{L}_{S+T}$ for a mini-batch data $\{(\boldsymbol{x}_i, \boldsymbol{y}_i)\}_m$ can be written as:

$$\mathcal{L}_{S+T} = -\frac{1}{\sum_i w_i} \sum_i w_i \cdot \text{CE}(f(\boldsymbol{x}_i), \boldsymbol{t}_i) + \frac{\beta}{m} \sum_i \text{KL}(f(\boldsymbol{x}_i), f(\widetilde{\boldsymbol{x}}_i)) \tag{10}$$

In addition, we provide the pseudo-code of the proposed Fast-LS-$l_0$, which incorporates SAT, TRADES and N-FGSM, in Algorithm 3.

# B  PROOFS

## B.1  PROOF OF LEMMA 3.2

*Proof.* Based on the definition of $\boldsymbol{\delta}_1$ and $\boldsymbol{\delta}_2$, we have $\mathcal{L}_\epsilon(\boldsymbol{x}, \boldsymbol{\theta}_1) = \mathcal{L}(\boldsymbol{x} + \boldsymbol{\delta}_1, \boldsymbol{\theta}_1)$ and $\mathcal{L}_\epsilon(\boldsymbol{x}, \boldsymbol{\theta}_2) = \mathcal{L}(\boldsymbol{x} + \boldsymbol{\delta}_2, \boldsymbol{\theta}_2)$. In this regard, we have:

$$\|\mathcal{L}_\epsilon(\boldsymbol{x}, \boldsymbol{\theta}_1) - \mathcal{L}_\epsilon(\boldsymbol{x}, \boldsymbol{\theta}_2)\| = \|\mathcal{L}(\boldsymbol{x} + \boldsymbol{\delta}_1, \boldsymbol{\theta}_1) - \mathcal{L}(\boldsymbol{x} + \boldsymbol{\delta}_2, \boldsymbol{\theta}_2)\| \tag{11}$$

When $\mathcal{L}(\boldsymbol{x} + \boldsymbol{\delta}_1, \boldsymbol{\theta}_1) \geq \mathcal{L}(\boldsymbol{x} + \boldsymbol{\delta}_2, \boldsymbol{\theta}_2)$ we have

$$\begin{aligned}
&\|\mathcal{L}(\boldsymbol{x} + \boldsymbol{\delta}_1, \boldsymbol{\theta}_1) - \mathcal{L}(\boldsymbol{x} + \boldsymbol{\delta}_2, \boldsymbol{\theta}_2)\| \\
=&\|\mathcal{L}(\boldsymbol{x} + \boldsymbol{\delta}_1, \boldsymbol{\theta}_1) - \mathcal{L}(\boldsymbol{x} + \boldsymbol{\delta}_1, \boldsymbol{\theta}_2) + \mathcal{L}(\boldsymbol{x} + \boldsymbol{\delta}_1, \boldsymbol{\theta}_2) - \mathcal{L}(\boldsymbol{x} + \boldsymbol{\delta}_2, \boldsymbol{\theta}_2)\| \\
\leq&\|\mathcal{L}(\boldsymbol{x} + \boldsymbol{\delta}_1, \boldsymbol{\theta}_1) - \mathcal{L}(\boldsymbol{x} + \boldsymbol{\delta}_1, \boldsymbol{\theta}_2)\|
\end{aligned} \tag{12}$$

The inequality above is derived from the optimality of $\delta_2$, which indicates $\mathcal{L}(\boldsymbol{x} + \boldsymbol{\delta}_1, \boldsymbol{\theta}_2) - \mathcal{L}(\boldsymbol{x} + \boldsymbol{\delta}_2, \boldsymbol{\theta}_2) \leq 0$ and the assumption $\mathcal{L}(\boldsymbol{x} + \boldsymbol{\delta}_1, \boldsymbol{\theta}_1) \geq \mathcal{L}(\boldsymbol{x} + \boldsymbol{\delta}_2, \boldsymbol{\theta}_2)$.

---

**Algorithm 3** Fast-LS-$l_0$

---

1: **Input:** Data: $\{(\boldsymbol{x}_i, \boldsymbol{y}_i)\}^n$; Initial target $\{\boldsymbol{t}_i\}^n = \{\boldsymbol{y}_i\}^n$; Batch size: $m$; Classifier: $f$; Enabling epoch: $E_s$; Momentum factor: $\alpha$; Balancing factor: $\beta$; TRADES mode: $mode$; Sparse level: $\epsilon$
2: **repeat**
3:     Fetch mini-batch data $\{(\boldsymbol{x}_i, \boldsymbol{t}_i)\}_m$ at current epoch $e$
4:     **for** $i = 1, ..., m$ **do**
5:         $\boldsymbol{\eta}_i \sim \mathcal{S}_{2\epsilon}(\boldsymbol{x}_i)$
6:         $\boldsymbol{x}_i = \boldsymbol{x}_i + \boldsymbol{\eta}_i$             // *Augment sample with additive noise*
7:         **if** $mode = F$ **then**
8:             $\widetilde{\boldsymbol{x}}_i = \max_{(\widetilde{\boldsymbol{x}}_i - \boldsymbol{x}_i) \in \mathcal{S}_\epsilon(\boldsymbol{x}_i)} \mathrm{KL}(f(\boldsymbol{x}_i), f(\widetilde{\boldsymbol{x}}_i))$
9:         **else if** $mode = T$ **then**
10:             $\widetilde{\boldsymbol{x}}_i = \max_{(\widetilde{\boldsymbol{x}}_i - \boldsymbol{x}_i) \in \mathcal{S}_\epsilon(\boldsymbol{x}_i)} \mathrm{CE}(f(\widetilde{\boldsymbol{x}}_i), \boldsymbol{t}_i)$
11:         **end if**
12:         $\boldsymbol{p}_i = \mathrm{softmax}(f(\boldsymbol{x}_i))$
13:         **if** $e > E_s$ **then**
14:             $\boldsymbol{t}_i = \alpha \times \boldsymbol{t}_i + (1 - \alpha) \times \boldsymbol{p}_i$
15:         **end if**
16:         $w_i = \max_j \boldsymbol{t}_{i,j}$
17:     **end for**
18:     Calculate $\mathcal{L}_{S+T}$ in Eq. (10)
19:     Update the parameters of $f$ on $\mathcal{L}_{S+T}$
20: **until** end of training

---

Similarly, when $\mathcal{L}(\boldsymbol{x} + \boldsymbol{\delta}_1, \boldsymbol{\theta}_1) \leq \mathcal{L}(\boldsymbol{x} + \boldsymbol{\delta}_2, \boldsymbol{\theta}_2)$ we have

$$
\begin{aligned}
&\|\mathcal{L}(\boldsymbol{x} + \boldsymbol{\delta}_1, \boldsymbol{\theta}_1) - \mathcal{L}(\boldsymbol{x} + \boldsymbol{\delta}_2, \boldsymbol{\theta}_2)\| \\
=&\|\mathcal{L}(\boldsymbol{x} + \boldsymbol{\delta}_1, \boldsymbol{\theta}_1) - \mathcal{L}(\boldsymbol{x} + \boldsymbol{\delta}_2, \boldsymbol{\theta}_1) + \mathcal{L}(\boldsymbol{x} + \boldsymbol{\delta}_2, \boldsymbol{\theta}_1) - \mathcal{L}(\boldsymbol{x} + \boldsymbol{\delta}_2, \boldsymbol{\theta}_2)\| \\
\leq&\|\mathcal{L}(\boldsymbol{x} + \boldsymbol{\delta}_2, \boldsymbol{\theta}_1) - \mathcal{L}(\boldsymbol{x} + \boldsymbol{\delta}_2, \boldsymbol{\theta}_2)\|
\end{aligned}
\tag{13}
$$

Without the loss of generality, we further bound $\|\mathcal{L}_\epsilon(\boldsymbol{x}, \boldsymbol{\theta}_1) - \mathcal{L}_\epsilon(\boldsymbol{x}, \boldsymbol{\theta}_2)\|$ based on (12). The derivation can be straightforwardly extended to (13) by replacing $\boldsymbol{\delta}_1$ with $\boldsymbol{\delta}_2$.

Based on the formulation of $\mathcal{L}$ in (1), $\|\mathcal{L}_\epsilon(\boldsymbol{x}, \boldsymbol{\theta}_1) - \mathcal{L}_\epsilon(\boldsymbol{x}, \boldsymbol{\theta}_2)\|$ can be further derived as follows:

$$
\begin{aligned}
\|\mathcal{L}_\epsilon(\boldsymbol{x}, \boldsymbol{\theta}_1) - \mathcal{L}_\epsilon(\boldsymbol{x}, \boldsymbol{\theta}_2)\| &\leq \left| \sum_{i \in \mathcal{S}_+} y_i \log \frac{h_i(\boldsymbol{x} + \boldsymbol{\delta}_1, \boldsymbol{\theta}_2)}{h_i(\boldsymbol{x} + \boldsymbol{\delta}_1, \boldsymbol{\theta}_1)} \right| \\
&= \sum_{i \in \mathcal{S}_+} y_i \left| \log \frac{1 + \sum_{j \neq i} \exp(f_j(\boldsymbol{x} + \boldsymbol{\delta}_1, \boldsymbol{\theta}_2) - f_i(\boldsymbol{x} + \boldsymbol{\delta}_1, \boldsymbol{\theta}_2))}{1 + \sum_{j \neq i} \exp(f_j(\boldsymbol{x} + \boldsymbol{\delta}_1, \boldsymbol{\theta}_1) - f_i(\boldsymbol{x} + \boldsymbol{\delta}_1, \boldsymbol{\theta}_1))} \right|
\end{aligned}
\tag{14}
$$

where $\mathcal{S}_+ = \{i \mid y_i > 0, h_i(\boldsymbol{x} + \boldsymbol{\delta}_1, \boldsymbol{\theta}_2) > h_i(\boldsymbol{x} + \boldsymbol{\delta}_1, \boldsymbol{\theta}_1)\}$. Then, according to the mediant inequality, we have

$$
\begin{aligned}
&\left| \log \frac{1 + \sum_{j \neq i} \exp(f_j(\boldsymbol{x} + \boldsymbol{\delta}_1, \boldsymbol{\theta}_2) - f_i(\boldsymbol{x} + \boldsymbol{\delta}_1, \boldsymbol{\theta}_2))}{1 + \sum_{j \neq i} \exp(f_j(\boldsymbol{x} + \boldsymbol{\delta}_1, \boldsymbol{\theta}_1) - f_i(\boldsymbol{x} + \boldsymbol{\delta}_1, \boldsymbol{\theta}_1))} \right| \\
\leq& \left| \log \frac{\sum_{j \neq i} \exp(f_j(\boldsymbol{x} + \boldsymbol{\delta}_1, \boldsymbol{\theta}_2) - f_i(\boldsymbol{x} + \boldsymbol{\delta}_1, \boldsymbol{\theta}_2))}{\sum_{j \neq i} \exp(f_j(\boldsymbol{x} + \boldsymbol{\delta}_1, \boldsymbol{\theta}_1) - f_i(\boldsymbol{x} + \boldsymbol{\delta}_1, \boldsymbol{\theta}_1))} \right| \\
\leq& \max_k \left| \log \frac{\exp(f_k(\boldsymbol{x} + \boldsymbol{\delta}_1, \boldsymbol{\theta}_2) - f_i(\boldsymbol{x} + \boldsymbol{\delta}_1, \boldsymbol{\theta}_2))}{\exp(f_k(\boldsymbol{x} + \boldsymbol{\delta}_1, \boldsymbol{\theta}_1) - f_i(\boldsymbol{x} + \boldsymbol{\delta}_1, \boldsymbol{\theta}_1))} \right| \\
\leq& \max_k |f_k(\boldsymbol{x} + \boldsymbol{\delta}_1, \boldsymbol{\theta}_2) - f_k(\boldsymbol{x} + \boldsymbol{\delta}_1, \boldsymbol{\theta}_1)| + |f_i(\boldsymbol{x} + \boldsymbol{\delta}_1, \boldsymbol{\theta}_2) - f_i(\boldsymbol{x} + \boldsymbol{\delta}_1, \boldsymbol{\theta}_1)| \\
\leq& 2L_{\boldsymbol{\theta}} \|\boldsymbol{\theta}_1 - \boldsymbol{\theta}_2\|
\end{aligned}
\tag{15}
$$

Note that the bound on the right of (15) is tight. The upper bound can be achieved asymptotically if the condition in (16) and the Lipschitz bound in Assumption 3.1 are satisfied.

$$
\left| |f_k(\boldsymbol{x} + \boldsymbol{\delta}_1, \boldsymbol{\theta}_2) - f_i(\boldsymbol{x} + \boldsymbol{\delta}_1, \boldsymbol{\theta}_2)| - |f_k(\boldsymbol{x} + \boldsymbol{\delta}_1, \boldsymbol{\theta}_1) - f_i(\boldsymbol{x} + \boldsymbol{\delta}_1, \boldsymbol{\theta}_1)| \right|
$$
$$
\gg \max_{j \neq k} \left| |f_j(\boldsymbol{x} + \boldsymbol{\delta}_1, \boldsymbol{\theta}_2) - f_i(\boldsymbol{x} + \boldsymbol{\delta}_1, \boldsymbol{\theta}_2)| - |f_j(\boldsymbol{x} + \boldsymbol{\delta}_1, \boldsymbol{\theta}_1) - f_i(\boldsymbol{x} + \boldsymbol{\delta}_1, \boldsymbol{\theta}_1)| \right| \tag{16}
$$

Combining (11)-(15), we have

$$
\|\mathcal{L}_\epsilon(\boldsymbol{x}, \boldsymbol{\theta}_1) - \mathcal{L}_\epsilon(\boldsymbol{x}, \boldsymbol{\theta}_2)\| \leq A_{\boldsymbol{\theta}} \|\boldsymbol{\theta}_1 - \boldsymbol{\theta}_2\|, \tag{17}
$$

where $A_{\boldsymbol{\theta}} = 2 \sum_{i \in \mathcal{S}_+} y_i L_{\boldsymbol{\theta}}$. $\qquad\square$

### B.2 PROOF OF LEMMA 3.4

*Proof.* Given (1), $\nabla_{\boldsymbol{\theta}} \mathcal{L}$ is computed as

$$
\begin{aligned}
\nabla_{\boldsymbol{\theta}} \mathcal{L}(\boldsymbol{x}, \boldsymbol{\theta}) &= - \sum_{i=0}^{K-1} y_i \left[ \nabla_{\boldsymbol{\theta}} f_i(\boldsymbol{x}, \boldsymbol{\theta}) - \frac{\sum_j \exp(f_j(\boldsymbol{x}, \boldsymbol{\theta})) \nabla_{\boldsymbol{\theta}} f_j(\boldsymbol{x}, \boldsymbol{\theta})}{\sum_j \exp(f_j(\boldsymbol{x}, \boldsymbol{\theta}))} \right] \\
&= \frac{\sum_j \exp(f_j(\boldsymbol{x}, \boldsymbol{\theta})) \nabla_{\boldsymbol{\theta}} f_j(\boldsymbol{x}, \boldsymbol{\theta})}{\sum_j \exp(f_j(\boldsymbol{x}, \boldsymbol{\theta}))} - \sum_{i=0}^{K-1} y_i \nabla_{\boldsymbol{\theta}} f_i(\boldsymbol{x}, \boldsymbol{\theta}) \\
&\overset{\text{def}}{=} \sum_{j=0}^{K-1} h_j(\boldsymbol{x}, \boldsymbol{\theta}) \nabla_{\boldsymbol{\theta}} f_j(\boldsymbol{x}, \boldsymbol{\theta}) - \sum_{i=0}^{K-1} y_i \nabla_{\boldsymbol{\theta}} f_i(\boldsymbol{x}, \boldsymbol{\theta})
\end{aligned} \tag{18}
$$

The second equality is based on the fact that $\{y_i\}_{i=0}^{K-1}$ is in a simplex. To simplify the notation, the last equation is based on the definition that $\{h_j\}_{j=0}^{K-1}$ is the result of softmax function applied to $\{f_j\}_{j=0}^{K-1}$, i.e., $h_j(\boldsymbol{x}, \boldsymbol{\theta}) = \frac{\exp(f_j(\boldsymbol{x}, \boldsymbol{\theta}))}{\sum_k \exp(f_k(\boldsymbol{x}, \boldsymbol{\theta}))}$. Therefore, we have $\sum_{j=0}^{K-1} h_j(\boldsymbol{x}, \boldsymbol{\theta}) = 1$ and $\forall j, h_j(\boldsymbol{x}, \boldsymbol{\theta}) > 0$.

According to the triangle inequality, we have:

$$
\begin{aligned}
&\|\nabla_{\boldsymbol{\theta}_1} \mathcal{L}(\boldsymbol{x} + \boldsymbol{\delta}_1, \boldsymbol{\theta}_1) - \nabla_{\boldsymbol{\theta}_2} \mathcal{L}(\boldsymbol{x} + \boldsymbol{\delta}_2, \boldsymbol{\theta}_2)\| \\
&\leq \|\nabla_{\boldsymbol{\theta}_1} \mathcal{L}(\boldsymbol{x} + \boldsymbol{\delta}_1, \boldsymbol{\theta}_1) - \nabla_{\boldsymbol{\theta}_1} \mathcal{L}(\boldsymbol{x} + \boldsymbol{\delta}_2, \boldsymbol{\theta}_1)\| + \|\nabla_{\boldsymbol{\theta}_1} \mathcal{L}(\boldsymbol{x} + \boldsymbol{\delta}_2, \boldsymbol{\theta}_1) - \nabla_{\boldsymbol{\theta}_2} \mathcal{L}(\boldsymbol{x} + \boldsymbol{\delta}_2, \boldsymbol{\theta}_2)\|
\end{aligned} \tag{19}
$$

Plug (18) to the first term on the right hand side of (19), we obtain:

$$
\|\nabla_{\boldsymbol{\theta}_1} \mathcal{L}(\boldsymbol{x} + \boldsymbol{\delta}_1, \boldsymbol{\theta}_1) - \nabla_{\boldsymbol{\theta}_1} \mathcal{L}(\boldsymbol{x} + \boldsymbol{\delta}_2, \boldsymbol{\theta}_1)\| \leq \sum_{i=0}^{K-1} y_i \|\nabla_{\boldsymbol{\theta}_1} f_i(\boldsymbol{x} + \boldsymbol{\delta}_1, \boldsymbol{\theta}_1) - \nabla_{\boldsymbol{\theta}_1} f_i(\boldsymbol{x} + \boldsymbol{\delta}_2, \boldsymbol{\theta}_1)\|
$$
$$
+ \left\| \sum_{j=0}^{K-1} h_j(\boldsymbol{x} + \boldsymbol{\delta}_1, \boldsymbol{\theta}_1) \nabla_{\boldsymbol{\theta}} f_j(\boldsymbol{x} + \boldsymbol{\delta}_1, \boldsymbol{\theta}_1) - \sum_{j=0}^{K-1} h_j(\boldsymbol{x} + \boldsymbol{\delta}_2, \boldsymbol{\theta}_1) \nabla_{\boldsymbol{\theta}} f_j(\boldsymbol{x} + \boldsymbol{\delta}_2, \boldsymbol{\theta}_1) \right\| \tag{20}
$$

The first term can be bounded based on Assumption 3.1. The second term can be bounded as follows:

$$\left\| \sum_{j=0}^{K-1} h_j(\boldsymbol{x}+\boldsymbol{\delta}_1, \boldsymbol{\theta}_1) \nabla_{\boldsymbol{\theta}} f_j(\boldsymbol{x}+\boldsymbol{\delta}_1, \boldsymbol{\theta}_1) - \sum_{j=0}^{K-1} h_j(\boldsymbol{x}+\boldsymbol{\delta}_2, \boldsymbol{\theta}_1) \nabla_{\boldsymbol{\theta}} f_j(\boldsymbol{x}+\boldsymbol{\delta}_2, \boldsymbol{\theta}_1) \right\|$$

$$\leq \left\| \sum_{j=0}^{K-1} h_j(\boldsymbol{x}+\boldsymbol{\delta}_1, \boldsymbol{\theta}_1) \nabla_{\boldsymbol{\theta}} f_j(\boldsymbol{x}+\boldsymbol{\delta}_1, \boldsymbol{\theta}_1) \right\| + \left\| \sum_{j=0}^{K-1} h_j(\boldsymbol{x}+\boldsymbol{\delta}_2, \boldsymbol{\theta}_1) \nabla_{\boldsymbol{\theta}} f_j(\boldsymbol{x}+\boldsymbol{\delta}_2, \boldsymbol{\theta}_1) \right\|$$

$$\leq \sum_{j=0}^{K-1} h_j(\boldsymbol{x}+\boldsymbol{\delta}_1, \boldsymbol{\theta}_1) \left\| \max_k \nabla_{\boldsymbol{\theta}} f_k(\boldsymbol{x}+\boldsymbol{\delta}_1, \boldsymbol{\theta}_1) \right\| + \sum_{j=0}^{K-1} h_j(\boldsymbol{x}+\boldsymbol{\delta}_2, \boldsymbol{\theta}_1) \left\| \max_k \nabla_{\boldsymbol{\theta}} f_k(\boldsymbol{x}+\boldsymbol{\delta}_2, \boldsymbol{\theta}_1) \right\|$$

$$\leq 2L_{\boldsymbol{\theta}} \tag{21}$$

Note that the bound on the right of (21) is tight. The first inequality is based on the triangle inequality. The second inequality and the third inequality can be achieved asymptotically when the equality of first-order Lipschitz continuity in Assumption 3.1 is achieved and the following condition is satisfied.

$$\exists k_1 \in \arg\max_i L_{\boldsymbol{\theta}}^{(i)}, h_{k_1}(\boldsymbol{x}+\boldsymbol{\delta}_1, \boldsymbol{\theta}_1) \to 1, \max_{j \neq k_1} h_j(\boldsymbol{x}+\boldsymbol{\delta}_1, \boldsymbol{\theta}_1) \to 0$$
$$\exists k_2 \in \arg\max_i L_{\boldsymbol{\theta}}^{(i)}, h_{k_2}(\boldsymbol{x}+\boldsymbol{\delta}_2, \boldsymbol{\theta}_1) \to 1, \max_{j \neq k_2} h_j(\boldsymbol{x}+\boldsymbol{\delta}_2, \boldsymbol{\theta}_1) \to 0 \tag{22}$$

Note that $k_1$ and $k_2$ are not always the same, since there may exist more than one biggest first-order Lipschitz constant.

Combining (20) and (21) together, we obtain:

$$\|\nabla_{\boldsymbol{\theta}_1} \mathcal{L}(\boldsymbol{x}+\boldsymbol{\delta}_1, \boldsymbol{\theta}_1) - \nabla_{\boldsymbol{\theta}_1} \mathcal{L}(\boldsymbol{x}+\boldsymbol{\delta}_2, \boldsymbol{\theta}_1)\| \leq 2L_{\boldsymbol{\theta}} + L_{\boldsymbol{\theta}\boldsymbol{x}}\|\boldsymbol{\delta}_2 - \boldsymbol{\delta}_1\| \tag{23}$$

Similarly, we have:

$$\|\nabla_{\boldsymbol{\theta}_1} \mathcal{L}(\boldsymbol{x}+\boldsymbol{\delta}_2, \boldsymbol{\theta}_1) - \nabla_{\boldsymbol{\theta}_2} \mathcal{L}(\boldsymbol{x}+\boldsymbol{\delta}_2, \boldsymbol{\theta}_2)\| \leq 2L_{\boldsymbol{\theta}} + L_{\boldsymbol{\theta}\boldsymbol{\theta}}\|\boldsymbol{\theta}_2 - \boldsymbol{\theta}_1\| \tag{24}$$

Combing the two inequalities above, we have:

$$\|\nabla_{\boldsymbol{\theta}} \mathcal{L}(\boldsymbol{x}+\boldsymbol{\delta}_1, \boldsymbol{\theta}_1) - \nabla_{\boldsymbol{\theta}} \mathcal{L}(\boldsymbol{x}+\boldsymbol{\delta}_2, \boldsymbol{\theta}_2)\| \leq A_{\boldsymbol{\theta}\boldsymbol{\theta}}\|\boldsymbol{\theta}_1 - \boldsymbol{\theta}_2\| + B_{\boldsymbol{\theta}\boldsymbol{\theta}} \tag{25}$$

where

$$A_{\boldsymbol{\theta}\boldsymbol{\theta}} = L_{\boldsymbol{\theta}\boldsymbol{\theta}}; \quad B_{\boldsymbol{\theta}\boldsymbol{\theta}} = 4L_{\boldsymbol{\theta}} + L_{\boldsymbol{\theta}\boldsymbol{x}}\|\boldsymbol{\delta}_1 - \boldsymbol{\delta}_2\| \tag{26}$$

$\square$

### B.3 PROOF OF THEOREM 4.1

*Proof.* For hard label $\boldsymbol{y}_h \in \{0,1\}^K$, let that the $j$-th elements of $\boldsymbol{y}_h$ be 1 and the rest be 0. By the definition of $A_{\boldsymbol{\theta}}$ in Lemma 3.2, we have

$$A_{\boldsymbol{\theta}}(\boldsymbol{y}_h) = 2L_{\boldsymbol{\theta}}. \tag{27}$$

It is known that $\sum_{i=0}^{K-1} h_i(\boldsymbol{x}, \boldsymbol{\theta}) = 1$, which means $\exists j, \ h_j(\boldsymbol{x}+\boldsymbol{\delta}_1, \boldsymbol{\theta}_2) \leq h_j(\boldsymbol{x}+\boldsymbol{\delta}_1, \boldsymbol{\theta}_1)$. Then, for soft label $\boldsymbol{y}_s \in (0,1)^K$, we have $|\mathcal{S}_+| < K$ where $\mathcal{S}_+ = \{i \mid y_i > 0, h_i(\boldsymbol{x}+\boldsymbol{\delta}_1, \boldsymbol{\theta}_2) > h_i(\boldsymbol{x}+\boldsymbol{\delta}_1, \boldsymbol{\theta}_1)\}$. Thus, it holds

$$A_{\boldsymbol{\theta}}(\boldsymbol{y}_s) = 2 \sum_{i \in \mathcal{S}_+} y_s^{(i)} L_{\boldsymbol{\theta}} \leq A_{\boldsymbol{\theta}}(\boldsymbol{y}_h). \tag{28}$$

The equality can be achieved asymptotically if $\sum_{i \notin \mathcal{S}_+} y_s^{(i)} \to 0$. $\square$

### B.4 PROOF OF THEOREM 4.2

*Proof.* By the definition of $\mathcal{L}_{\epsilon,\alpha}$ in (8), we have

$$
\begin{aligned}
&\|\nabla_{\boldsymbol{\theta}_1}\mathcal{L}_{\epsilon,\alpha}(\boldsymbol{x},\boldsymbol{\theta}_1) - \nabla_{\boldsymbol{\theta}_2}\mathcal{L}_{\epsilon,\alpha}(\boldsymbol{x},\boldsymbol{\theta}_2)\| \\
&\leq (1-\alpha)\|\nabla_{\boldsymbol{\theta}_1}\mathcal{L}(\boldsymbol{x},\boldsymbol{\theta}_1) - \nabla_{\boldsymbol{\theta}_1}\mathcal{L}(\boldsymbol{x},\boldsymbol{\theta}_2)\| + \alpha\|\nabla_{\boldsymbol{\theta}_1}\mathcal{L}_\epsilon(\boldsymbol{x},\boldsymbol{\theta}_1) - \nabla_{\boldsymbol{\theta}_1}\mathcal{L}_\epsilon(\boldsymbol{x},\boldsymbol{\theta}_2)\|
\end{aligned}
\tag{29}
$$

According to (24) in the proof of Lemma 3.4, the first term of the right hand side of (29) can be derived as

$$
\|\nabla_{\boldsymbol{\theta}_1}\mathcal{L}(\boldsymbol{x},\boldsymbol{\theta}_1) - \nabla_{\boldsymbol{\theta}_2}\mathcal{L}(\boldsymbol{x},\boldsymbol{\theta}_2)\| \leq L_{\boldsymbol{\theta\theta}}\|\boldsymbol{\theta}_1 - \boldsymbol{\theta}_2\| + 2L_{\boldsymbol{\theta}}.
\tag{30}
$$

According to Lemma 3.4, the second term of the right hand side of (29) satisifies

$$
\|\nabla_{\boldsymbol{\theta}_1}\mathcal{L}_\epsilon(\boldsymbol{x},\boldsymbol{\theta}_1) - \nabla_{\boldsymbol{\theta}_2}\mathcal{L}_\epsilon(\boldsymbol{x},\boldsymbol{\theta}_2)\| \leq L_{\boldsymbol{\theta\theta}}\|\boldsymbol{\theta}_1 - \boldsymbol{\theta}_2\| + L_{\boldsymbol{\theta x}}\|\boldsymbol{\delta}_1 - \boldsymbol{\delta}_2\| + 4L_{\boldsymbol{\theta}}.
\tag{31}
$$

Combining (29), (30) and (31), we have

$$
\|\nabla_{\boldsymbol{\theta}_1}\mathcal{L}_{\epsilon,\alpha}(\boldsymbol{x},\boldsymbol{\theta}_1) - \nabla_{\boldsymbol{\theta}_2}\mathcal{L}_{\epsilon,\alpha}(\boldsymbol{x},\boldsymbol{\theta}_2)\| \leq A_{\boldsymbol{\theta\theta}}\|\boldsymbol{\theta}_1 - \boldsymbol{\theta}_2\| + B'_{\boldsymbol{\theta\delta}},
\tag{32}
$$

where $A_{\boldsymbol{\theta\theta}} = L_{\boldsymbol{\theta\theta}}$ and $B'_{\boldsymbol{\theta\delta}} = \alpha L_{\boldsymbol{\theta x}}\|\boldsymbol{\delta}_1 - \boldsymbol{\delta}_2\| + 2(1+\alpha)L_{\boldsymbol{\theta}}$. $\qquad\square$

## C THEORETICAL ANALYSIS OF ReLU NETWORKS

Similar to Liu et al. (2020), we first make the following assumptions for the functions $\{f_i\}_{i=0}^{K-1}$ represented by a ReLU network.

**Assumption C.1.** $\forall i \in \{0, 1, ..., K-1\}$, the function $f_i$ satisfies the following conditions:

$$
\forall \boldsymbol{x}, \boldsymbol{\theta}_1, \boldsymbol{\theta}_2, \quad \|f_i(\boldsymbol{x},\boldsymbol{\theta}_1) - f_i(\boldsymbol{x},\boldsymbol{\theta}_2)\| \leq L_{\boldsymbol{\theta}}\|\boldsymbol{\theta}_1 - \boldsymbol{\theta}_2\|,
\tag{33}
$$

$$
\forall \boldsymbol{\theta}, \boldsymbol{x}_1, \boldsymbol{x}_2, \quad \|f_i(\boldsymbol{x}_1,\boldsymbol{\theta}) - f_i(\boldsymbol{x}_2,\boldsymbol{\theta})\| \leq L_{\boldsymbol{x}}\|\boldsymbol{x}_1 - \boldsymbol{x}_2\|,
\tag{34}
$$

$$
\forall \boldsymbol{x}, \boldsymbol{\theta}_1, \boldsymbol{\theta}_2, \quad \|\nabla_{\boldsymbol{\theta}}f_i(\boldsymbol{x},\boldsymbol{\theta}_1) - \nabla_{\boldsymbol{\theta}}f_i(\boldsymbol{x},\boldsymbol{\theta}_2)\| \leq L_{\boldsymbol{\theta\theta}}\|\boldsymbol{\theta}_1 - \boldsymbol{\theta}_2\| + C_{\boldsymbol{\theta\theta}},
\tag{35}
$$

$$
\forall \boldsymbol{\theta}, \boldsymbol{x}_1, \boldsymbol{x}_2, \quad \|\nabla_{\boldsymbol{\theta}}f_i(\boldsymbol{x}_1,\boldsymbol{\theta}) - \nabla_{\boldsymbol{\theta}}f_i(\boldsymbol{x}_2,\boldsymbol{\theta})\| \leq L_{\boldsymbol{\theta x}}\|\boldsymbol{x}_1 - \boldsymbol{x}_2\| + C_{\boldsymbol{\theta x}}.
\tag{36}
$$

Compared to Assumption 3.1 and 3.3, we modify the the second-order smoothness assumptions by adding two constants $C_{\boldsymbol{\theta\theta}}$ and $C_{\boldsymbol{\theta x}}$, respectively. They denote the upper bound of the gradient difference in the neighborhood at non-smooth point. Thus, they quantify how drastically the (sub)gradients can change in a sufficiently small region in the parameter space.

Based on Assumption C.1, we have the following corollary:

**Corollary C.2.** *If Assumption C.1 is satisfied, it holds*

$$
\|\mathcal{L}_\epsilon(\boldsymbol{x},\boldsymbol{\theta}_1) - \mathcal{L}_\epsilon(\boldsymbol{x},\boldsymbol{\theta}_2)\| \leq A_{\boldsymbol{\theta}}\|\boldsymbol{\theta}_1 - \boldsymbol{\theta}_2\|,
\tag{37}
$$

$$
\|\nabla_{\boldsymbol{\theta}}\mathcal{L}_\epsilon(\boldsymbol{x},\boldsymbol{\theta}_1) - \nabla_{\boldsymbol{\theta}}\mathcal{L}_\epsilon(\boldsymbol{x},\boldsymbol{\theta}_2)\| \leq A_{\boldsymbol{\theta\theta}}\|\boldsymbol{\theta}_1 - \boldsymbol{\theta}_2\| + B_{\boldsymbol{\theta\delta}} + C_{\boldsymbol{\theta\theta}} + C_{\boldsymbol{\theta x}}.
\tag{38}
$$

*The Lipschitz constant $A_{\boldsymbol{\theta}} = 2\sum_{i\in\mathcal{S}_+}y_iL_{\boldsymbol{\theta}}$, $A_{\boldsymbol{\theta\theta}} = L_{\boldsymbol{\theta\theta}}$ and $B_{\boldsymbol{\theta\delta}} = L_{\boldsymbol{\theta x}}\|\boldsymbol{\delta}_1 - \boldsymbol{\delta}_2\| + 4L_{\boldsymbol{\theta}}$ where $\boldsymbol{\delta}_1 \in \arg\max_{\boldsymbol{\delta}\in\mathcal{S}_\epsilon}\mathcal{L}(\boldsymbol{x}+\boldsymbol{\delta},\boldsymbol{\theta}_1)$ and $\boldsymbol{\delta}_2 \in \arg\max_{\boldsymbol{\delta}\in\mathcal{S}_\epsilon}\mathcal{L}(\boldsymbol{x}+\boldsymbol{\delta},\boldsymbol{\theta}_2)$.*

The proof is similar to that of Lemma 3.2 and 3.4. Corollary C.2 indicates a more craggy loss landscape in the adversarial training of networks with non-smooth activations.

Additionally, the Theorem 4.2 can be easily extended to accommodate Assumption C.1.

**Corollary C.3.** *If Assumption C.1 holds, then we have*

$$
\|\nabla_{\boldsymbol{\theta}}\mathcal{L}_{\epsilon,\alpha}(\boldsymbol{x},\boldsymbol{\theta}_1) - \nabla_{\boldsymbol{\theta}}\mathcal{L}(\boldsymbol{x},\boldsymbol{\theta}_2)\| \leq A_{\boldsymbol{\theta\theta}}\|\boldsymbol{\theta}_1 - \boldsymbol{\theta}_2\| + B'_{\boldsymbol{\theta\delta}} + C_{\boldsymbol{\theta\theta}} + C_{\boldsymbol{\theta x}}.
\tag{39}
$$

*The Lipschitz constant $A_{\boldsymbol{\theta\theta}} = L_{\boldsymbol{\theta\theta}}$ and $B'_{\boldsymbol{\theta\delta}} = \alpha L_{\boldsymbol{\theta x}}\|\boldsymbol{\delta}_1 - \boldsymbol{\delta}_2\| + 2(1+\alpha)L_{\boldsymbol{\theta}}$ where $\boldsymbol{\delta}_1 \in \arg\max_{\boldsymbol{\delta}\in\mathcal{S}_\epsilon}\mathcal{L}(\boldsymbol{x}+\boldsymbol{\delta},\boldsymbol{\theta}_1)$ and $\boldsymbol{\delta}_2 \in \arg\max_{\boldsymbol{\delta}\in\mathcal{S}_\epsilon}\mathcal{L}(\boldsymbol{x}+\boldsymbol{\delta},\boldsymbol{\theta}_2)$.*

## D DISCUSSION OF THE UPPER BOUND OF $\|\boldsymbol{\delta}_1 - \boldsymbol{\delta}_2\|$

We define the $l_p$ adversarial budget for the perturbation $\boldsymbol{\delta} \in \mathbb{R}^d$ as $\mathcal{S}_\epsilon^{(p)} = \{\boldsymbol{\delta} \mid \|\boldsymbol{\delta}\|_p \leq \epsilon, 0 \leq \boldsymbol{x} + \boldsymbol{\delta} \leq 1\}$. Therefore, we have $\|\boldsymbol{\delta}_1 - \boldsymbol{\delta}_2\|_p \leq 2\epsilon$, and $\forall i, 0 \leq |\delta_1^{(i)} - \delta_2^{(i)}| \leq 1$ where $\delta_1^{(i)}$ and

$\delta_2^{(i)}$ are the $i$-th element of $\boldsymbol{\delta}_1$ and $\boldsymbol{\delta}_2$, respectively. For convenience, we denote $\boldsymbol{\delta}_1 - \boldsymbol{\delta}_2$ as $\Delta\boldsymbol{\delta}$ and $\delta_1^{(i)} - \delta_2^{(i)}$ as $\Delta\delta_i$ in the following.

Assume that $\epsilon \ll d$ for $l_0$, $l_1$ and $l_2$ bounded perturbations, and $\epsilon \ll 1$ for the $l_\infty$ bounded perturbation. Then, $\forall q \geq 1$, we have

$$
\begin{aligned}
l_0 \text{ budget:} \quad & \sum_i |\Delta\delta_i|^q \leq 2\epsilon, \\
l_1 \text{ budget:} \quad & \sum_i |\Delta\delta_i|^q \leq D_1 + (2\epsilon - D_1)^q, \\
l_2 \text{ budget:} \quad & \sum_i |\Delta\delta_i|^q \leq D_2 + (4\epsilon^2 - D_2)^{\frac{q}{2}}, \\
l_\infty \text{ budget:} \quad & \sum_i |\Delta\delta_i|^q \leq d \times (2\epsilon)^q,
\end{aligned}
\tag{40}
$$

where $D_1 = \lfloor 2\epsilon \rceil$ and $D_2 = \lfloor 4\epsilon^2 \rceil$. The derived upper bounds are tight because

**(1) $l_0$ budget:** The equality achieves when the location of non-zero elements in $\boldsymbol{\delta}_1$ and $\boldsymbol{\delta}_2$ has no overlap, and the magnitude of their non-zero elements reaches $\pm 1$.

**(2) $l_1$ budget:** Since $0 \leq |\Delta\delta_i| \leq 1$, the equality achieves when there exists at most one $\Delta\delta_k$ such that $|\Delta\delta_k| < 1$ and $\forall j \neq k$, $|\Delta\delta_j| = 1$. The maximum number of $\Delta\delta_j$ is $\lfloor 2\epsilon \rfloor$. Then, according to $\|\Delta\boldsymbol{\delta}\|_1 \leq 2\epsilon$, we have $|\Delta\delta_k| = 2\epsilon - 1 \times \lfloor 2\epsilon \rfloor$.

**(3) $l_2$ budget:** The derivation is similar to that of the $l_1$ case.

**(4) $l_\infty$ budget:** The equality achieves when $\boldsymbol{\delta}_1 = -\boldsymbol{\delta}_2$.

On popular benchmark CIFAR-10, $d = 32 \times 32 \times 3 = 3072$, and the commonly used values of $\epsilon$ in the $l_0$, $l_1$, $l_2$ and $l_\infty$ cases are 360, 24, 0.5 and $8/255$, respectively (Madry et al., 2018; Zhong et al., 2024; Croce & Hein, 2021; Jiang et al., 2023). Substitute these into (40), we can easily get that $\forall q \geq 1$, the upper bound of $\sum_i |\Delta\delta_i|^q$ is significantly larger in the $l_0$ case than the other cases. For instance, $(2\epsilon - D_1)^q$, $(4\epsilon^2 - D_2)^{\frac{q}{2}}$ and $(2\epsilon)^q$ reach their respective maximum values when $q = 1$, since all of them are smaller than 1. Then, the upper bounds of $\sum_i |\Delta\delta_i|^1$ in the $l_0$, $l_1$, $l_2$ and $l_\infty$ cases are 720, 24, 1 and $49152/255 \approx 192.8$, respectively.

Furthermore, the $l_q$ norm of $\Delta\boldsymbol{\delta}$ is defined as follows:

$$
\|\Delta\boldsymbol{\delta}\|_q = \left( \sum_i |\Delta\delta_i|^q \right)^{\frac{1}{q}}.
\tag{41}
$$

Since the upper bound of $\sum_i |\Delta\delta_i|^q$ in the $l_0$ case is larger than 1 for all $q \geq 1$, we can also derive that $\forall q \geq 1$, the upper bound of $\|\Delta\boldsymbol{\delta}\|_q$ is always significantly larger in the $l_0$ case than the other cases.

# E   MORE EXPERIMENTAL DETAILS

## E.1   LOCATION DIFFERENCE BETWEEN ADVERSARIAL EXAMPLES GENERATED BY 1-STEP sPGD AND sAA

As illustrated in Figure 4, the adversarial perturbations generated by one-step sPGD during training are almost completely different from those generated by sAA in location rather than magnitude. Combining with the results in Table 2, we can demonstrate that CO in $l_0$ adversarial training is primarily due to sub-optimal perturbation locations rather than magnitudes.

## E.2   DISTANCES BETWEEN GRADIENTS INDUCED BY 1-STEP AND MULTI-STEP ATTACKS

Based on the Lipschitz smoothness assumption in Inequality (6), the gradient difference arising from approximated adversarial perturbations is bounded by $L_{\boldsymbol{\theta x}}\|\boldsymbol{\delta}_1 - \boldsymbol{\delta}_2\|$ where $\boldsymbol{\delta}_1$ is the perturbation generated by 1-step attack and $\boldsymbol{\delta}_2$ is the optimal perturbation. Based on the same reason that $l_0$ norm

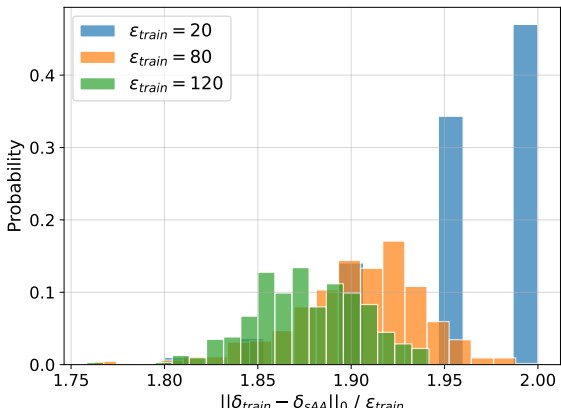

Figure 4: The distribution of the normalized $l_0$ distance between training adversarial examples generated by 1-step sPGD and sAA. The models trained on 1-step sAT with different training $\epsilon$ are evaluated.

Table 5: Average $l_2$ distances between gradients induced by 1-step and multi-step attacks, represented by $\|\nabla_{\boldsymbol{\theta}}\mathcal{L}_\epsilon(\boldsymbol{x} + \delta_{one}) - \nabla_{\boldsymbol{\theta}}\mathcal{L}_\epsilon(\boldsymbol{x} + \delta_{multi})\|_2$. The gradients are calculated of the training set of CIFAR-10 (Krizhevsky et al., 2009). The $l_0, l_1, l_2$ and $l_\infty$ models are obtained by 1-step sAT (Zhong et al., 2024), Fast-EG-$l_1$ (Jiang et al., 2023), 1-step PGD (Rice et al., 2020) and GradAlign (Andriushchenko et al., 2020), respectively. The 1-step and multi-step $l_0$ attacks are 1-step and 10000-step sPGD (Zhong et al., 2024), respectively. The 1-step and multi-step $l_1$ attacks are 1-step Fast-EG-$l_1$ and 100-step APGD (Croce & Hein, 2021), respectively. The 1-step and multi-step attacks for other norms are 1-step PGD (Madry et al., 2018) and 100-step APGD (Croce & Hein, 2020b), respectively.

| Model | $l_0$ ($\epsilon = 1$) | $l_1$ ($\epsilon = 24$) | $l_2$ ($\epsilon = 0.5$) | $l_\infty$ ($\epsilon = 8/255$) |
|---|---|---|---|---|
| $l_2$ distance | 15.8 | $9.1 \times 10^{-4}$ | $3.6 \times 10^{-4}$ | $6.7 \times 10^{-4}$ |

is not a proper norm, $\|\boldsymbol{\delta}_1 - \boldsymbol{\delta}_2\|$ is significantly larger in $l_0$ cases than $l_\infty$, $l_2$ and $l_1$ cases, which makes 1-step adversarial training more challenging in $l_0$ cases. To corroborate this, we compare the distance between gradients induced by 1-step and multi-step attacks. As presented in Table 5, the average distance between gradients induced by 1-step and multi-step $l_0$ attacks is 5 orders of magnitude greater than those in the $l_1$, $l_2$ and $l_\infty$ cases, even when a single pixel is perturbed. This finding indicates that the loss landscape of $l_0$ adversarial training is significantly more craggy than other cases in the input space.

### E.3 COMPARISON WITH OTHER BASELINES

Table 6: Comparison with other baselines in robust accuracy (%) by sAA. The target sparsity level $\epsilon = 20$. We compare PreAct ResNet-18 (He et al., 2016a) models trained on CIFAR-10 (Krizhevsky et al., 2009) with 100 epochs. The *italic numbers* indicate catastrophic overfitting (CO) happens.

| Method | ATTA | ATTA + S&N | GA | GA + S&N | Fast-BAT | FLC Pool | N-AAER |
|---|---|---|---|---|---|---|---|
| Robust Acc. | *0.0* | 54.7 | *0.0* | 34.4 | 14.1 | *0.0* | *0.1* |

| Method | N-LAP | LS | NuAT | AdvLC | MART | Ours + AWP | **Ours** |
|---|---|---|---|---|---|---|---|
| Robust Acc. | *0.0* | *0.0* | 51.9 | 47.6 | 48.0 | 47.2 | **63.0** |

In this section, we undertake a more comprehensive comparison between our proposed Fast-LS-$l_0$ and other baselines (ATTA (Zheng et al., 2019), GradAlign (GA) (Andriushchenko & Flammarion, 2020), Fast-BAT (Zhang et al., 2022), N-AAER (Lin et al., 2024b), N-LAP (Lin et al., 2024a), label smoothing (LS) (Szegedy et al., 2016), NuAT (Sriramanan et al., 2021), AdvLC (Li & Spratling, 2023), MART (Wang et al., 2020) and AWP Wu et al. (2020)), which either claim to mitigate catastrophic overfitting or claim to incorporate different smoothing techniques.

As demonstrated in Table 6, our method achieves the strongest robustness against sAA. First, naive LS turns out ineffective under the $l_0$ setting. The performance of Fast-BAT, NuAT, AdvLC and

MART is not as good as the method we use. Second, FLC Pool, N-AAER, N-LAP, ATTA and GradAlign suffer from CO, since they incorporate neither soft labels nor trade-off loss function. Combining ATTA and GradAlign with SAT and N-FGSM, which introduces soft labels, can effectively mitigate CO, but these settings still underperform our method by a large margin. Finally, although AWP can find a flatter minimum, it requires dedicated hyperparameters tuning and introduces additional overhead. Under its default HP setting, AWP results in a deterioration of performance in the $l_0$ case.

## E.4 MORE RESULTS OF SECTION 5.2

Table 7: Robust accuracy (%) of various models on different attacks that generate $l_0$ bounded perturbations, where the sparsity level $\epsilon = 10$. The models are PreAct ResNet-18 trained on **CIFAR-100** (Krizhevsky et al., 2009) with $\epsilon = 60$. Note that the results of vanilla sAT and sTRADES are obtained from (Zhong et al., 2024), CornerSearch (CS) is evaluated on 1000 samples due to its high computational complexity.

| Model | Time Cost | Clean | Black-Box | | White-Box | | | sAA |
|---|---|---|---|---|---|---|---|---|
| | | | CS | RS | SAIF | $\text{sPGD}_{\text{proj}}$ | $\text{sPGD}_{\text{unproj}}$ | |
| *Multi-step* | | | | | | | | |
| sAT | 4h 27m | 67.0 | 44.3 | 41.6 | 60.9 | 56.8 | 58.0 | 41.6 |
| **+S** | 5h 02m | 65.5 | 50.8 | 50.7 | 61.4 | 59.2 | 60.5 | 50.7 |
| **+S&N** | 4h 58m | 64.3 | 53.0 | 52.9 | 61.2 | 59.2 | 59.6 | 52.8 |
| sTRADES | 5h 10m | 70.9 | 52.8 | 50.3 | 65.2 | 64.0 | 63.7 | 50.2 |
| **+S** | 5h 53m | 65.1 | 54.9 | 54.6 | 62.7 | 61.0 | 60.5 | 54.6 |
| **+S&N** | 5h 40m | 63.8 | 56.5 | 55.6 | 61.2 | 60.5 | 59.0 | **55.3** |
| *One-step* | | | | | | | | |
| **Fast-LS-$l_0$ (T)** | 1h 05m | 65.3 | 54.5 | 54.3 | 60.4 | 55.6 | 54.4 | 52.2 |
| **Fast-LS-$l_0$ (F)** | 1h 26m | 65.0 | 56.2 | 54.6 | 60.8 | 54.9 | 54.9 | **52.3** |

Table 8: Robust accuracy (%) of various models on different attacks that generate $l_0$ bounded perturbations, where the sparsity level $\epsilon = 12$. The models are PreAct ResNet-18 trained on **GTSRB** (Stallkamp et al., 2012) with $\epsilon = 72$. All methods are evaluated on 500 samples, and CornerSearch (CS) is not evaluated here due to its high computational complexity.

| Model | Time Cost | Clean | Black-Box | | White-Box | | | sAA |
|---|---|---|---|---|---|---|---|---|
| | | | CS | RS | SAIF | $\text{sPGD}_{\text{proj}}$ | $\text{sPGD}_{\text{unproj}}$ | |
| *Multi-step* | | | | | | | | |
| sAT | 1h 3m | 98.4 | - | 43.2 | 92.4 | 96.0 | 96.2 | 43.2 |
| **+S** | 1h 3m | 98.6 | - | 75.6 | 97.2 | 97.0 | 96.4 | 75.6 |
| **+S&N** | 1h 2m | 98.4 | - | 77.8 | 97.4 | 96.8 | 95.4 | **77.6** |
| sTRADES | 1h 6m | 97.8 | - | 67.6 | 94.0 | 95.6 | 95.0 | 67.4 |
| **+S** | 1h 5m | 96.8 | - | 76.4 | 94.6 | 94.4 | 92.6 | 76.4 |
| **+S&N** | 1h 7m | 95.6 | - | 75.4 | 93.6 | 92.6 | 91.2 | 75.2 |
| *One-step* | | | | | | | | |
| **Fast-LS-$l_0$ (T)** | 7m | 97.8 | - | 75.2 | 89.2 | 74.4 | 74.4 | 63.2 |
| **Fast-LS-$l_0$ (F)** | 9m | 98.6 | - | 80.4 | 94.2 | 75.0 | 79.8 | **67.8** |

The results on CIFAR-100 and GTSRB datasets are presented in Table 7 and 8, respectively. The findings are consistent with those observed in Table 4(a), further validating the effectiveness of the proposed methods across different datasets. In contrast to the settings in (Zhong et al., 2024), we resize the images in GTSRB to $32 \times 32$ instead of $224 \times 224$ and retrain the models from scratch. The model are trained with $\epsilon = 72$ and evaluated for robustness with $\epsilon = 12$. It is important to note that due to the smaller search space resulting from low-resolution images, the attack success rate of the black-box Sparse-RS (RS) under this setting is significantly higher than that reported in (Zhong et al., 2024).

Table 9: Average robust accuracy against sAA (Zhong et al., 2024) obtained from three runs, where the sparsity level $\epsilon = 20$. The variances are shown in brackets. The configurations are the same as in Table 4(a). Note that we do not include the results of vanilla sAT and sTRADES since their results are obtained from (Zhong et al., 2024).

| Model | 20-step sAT | | 20-step sTRADES | | Fast-LS-$l_0$ (T) | Fast-LS-$l_0$ (F) |
| | + S | + S&N | + S | + S&N | | |
|---|---|---|---|---|---|---|
| Acc. | 55.5 ($\pm$ 1.3) | 61.2 ($\pm$ 0.2) | 64.1 ($\pm$ 0.9) | 65.5 ($\pm$ 0.7) | 63.0 ($\pm$ 0.7) | 62.1 ($\pm$ 0.6) |

### E.5 STANDARD DEVIATION OF ROBUST ACCURACY AGAINST SPARSE-AUTOATTACK OF TABLE 4(A)

To better validate the effectiveness of our method, we report the standard deviations of robust accuracy against sAA in Table 9. We calculate these standard deviations by running the experiments three times with different random seeds. The configurations are the same as in Table 4(a). It can be observed that the fluctuation introduced by different random seeds does not outweigh the performance gain from the evaluated approaches.

### E.6 EVALUATION ON DIFFERENT NETWORKS

Table 10: Robust accuracy (%) of various networks against sAA on CIFASR-10, where the sparsity level $\epsilon = 20$. The networks are adversarially trained with different methods, including 1-step sAT, 1-step sTRADES and the proposed Fast-LS-$l_0$.

| | PRN-18 | ConvNeXt-T | Swin-T |
|---|---|---|---|
| 1-step sAT | 0.0 | 0.8 | 0.1 |
| 1-step sTRADES | 31.0 | 71.0 | 43.2 |
| **Fast-LS-$l_0$** | **63.0** | **78.6** | **58.9** |

Despite the effectiveness of our method on PreActResNet-18 (PRN-18) and ResNet-34, the performance of our Fast-LS-$l_0$ and its ablations on different networks remains unexplored. In this regard, we further evaluate our method on two popular architectures, i.e., ConvNeXt (Liu et al., 2022) and Swin Transformer (Liu et al., 2021b). Note that we adopt their tiny versions for CIFAR-10, which have a similar number of parameters as ResNet-18, and we follow the training settings of their CIFAR-10 implementations. The other experimental settings are the same as those described in Section 5.1. As shown in Table 10, vanilla adversarial training results in CO on all networks, and our method produces the best robust accuracy against sAA, demonstrating the effectiveness of our method on different networks. Notably, ConvNeXt shows surprisingly strong robustness against sAA, suggesting that advanced architecture design and dedicated hyperparameter tuning can provide additional performance gains. However, as Transformers has struggled to perform well on small datasets without pretraining (Debenedetti et al., 2023), Swin Transformer also underperforms CNN-based networks in this scenario.

### E.7 LOSS LANDSCAPE OF ONE-STEP SAT WITH DIFFERENT $\epsilon$

As supplementary of Figure 2, we visualize the loss landscapes of 1-step sAT (Zhong et al., 2024) with different $\epsilon$, including 20, 40 and 120, in Figure 5. It can be observed that the $l_0$ adversarial loss exhibits a drastic increase in response to relatively minor alterations in the $\theta$-space. Moreover, the degree of non-smoothness increases in proportion to $\epsilon$, which is consistent with the observation in Figure 2 (a).

### E.8 SMOOTHER LOSS LANDSCAPE INDUCED BY SOFT LABEL AND TRADE-OFF LOSS FUNCTION

The effectiveness of soft label and trade-off loss function in improving the performance of $l_0$ adversarial training is demonstrated in Section 5.1 and 5.2. Additionally, we visualize the curves of top-10 eigenvalues of Hessian matrices of the different methods discussed in Section 5.1 and their respective loss landscapes in Figure 6. Note that since N-FGSM results in a larger upper bound of $\|\delta_1 - \delta_2\|$, it is not considered here to make a fair comparison. Figure 6 (a) shows that sTRADES

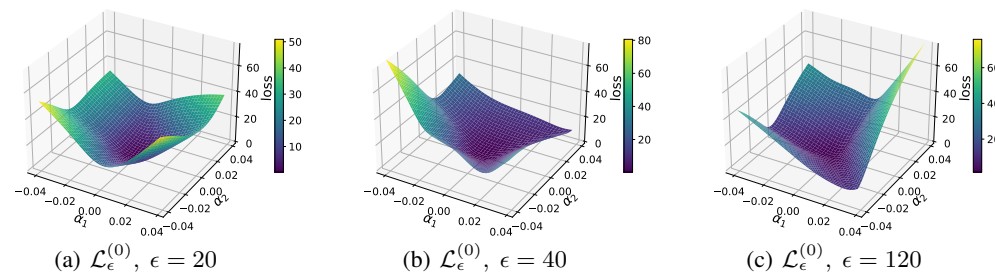

(a) $\mathcal{L}_\epsilon^{(0)}$, $\epsilon = 20$     (b) $\mathcal{L}_\epsilon^{(0)}$, $\epsilon = 40$     (c) $\mathcal{L}_\epsilon^{(0)}$, $\epsilon = 120$

Figure 5: Loss landscape of 1-step sAT (Zhong et al., 2024) with different $\epsilon$ values on the training set of CIFAR-10 (Krizhevsky et al., 2009). The architecture of the model is PreactResNet-18. **(a)** Landscape of $\mathcal{L}_\epsilon^{(0)}(\boldsymbol{x}, \boldsymbol{\theta} + \alpha_1 \boldsymbol{v}_1 + \alpha_2 \boldsymbol{v}_2)$ with $\epsilon = 20$, where $\boldsymbol{v}_1$ and $\boldsymbol{v}_2$ are the eigenvectors corresponding to the top 2 eigenvalues of the Hessian matrices, respectively. **(b)** Landscape of $\mathcal{L}_\epsilon^{(0)}$ with $\epsilon = 40$. **(c)** Landscape of $\mathcal{L}_\epsilon^{(0)}$ with $\epsilon = 120$.

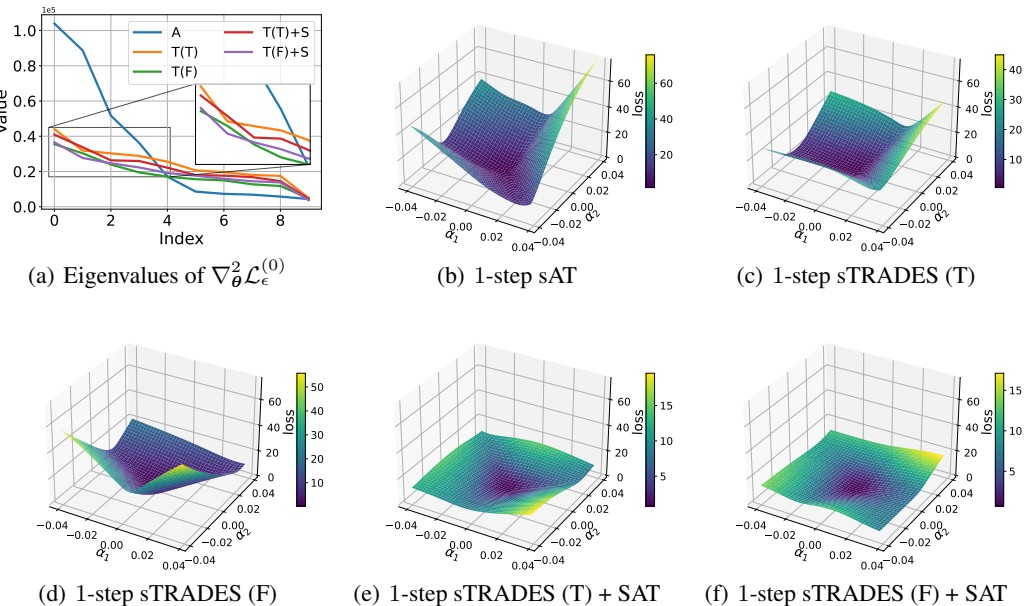

(a) Eigenvalues of $\nabla_{\boldsymbol{\theta}}^2 \mathcal{L}_\epsilon^{(0)}$    (b) 1-step sAT    (c) 1-step sTRADES (T)

(d) 1-step sTRADES (F)    (e) 1-step sTRADES (T) + SAT    (f) 1-step sTRADES (F) + SAT

Figure 6: Smoothness visualization of different methods with $\epsilon = 120$ on the training set of CIFAR-10 (Krizhevsky et al., 2009). The architecture of the model is PreactResNet-18. **(a)** Top-10 eigenvalues of $\nabla_{\boldsymbol{\theta}}^2 \mathcal{L}_\epsilon^{(0)}(\boldsymbol{x}, \boldsymbol{\theta})$ of different methods. A and T denote 1-step sAT and 1-step sTRADES, respectively. T and F in the brackets are two respective versions of sTRADES indicated in Sec. 5.1. **(b)** Loss landscape of 1-step sAT. **(c)** Loss landscape of 1-step sTRADES (T). **(d)** Loss landscape of 1-step sTRADES (F). **(e)** Loss landscape of 1-step sTRADES (T) + SAT. **(f)** Loss landscape of 1-step sTRADES (F) + SAT.

induces considerably smaller eigenvalues of Hessian matrices compared to sAT, while the difference between sTRADES (T) and sTRADES (F) is negligible. SAT, on the other hand, has only a marginal effect on the eigenvalues. However, as illustrated in Figure 6 (b)-(f), SAT plays a crucial role in smoothing the loss landscape, which relates to the change rate of loss, i.e., the first-order smoothness. These observations align with the theoretical derivation presented in Section 4, indicating that soft label improves the first-order smoothness, while trade-off loss function contributes to the second-order smoothness.

### E.9 ABLATION STUDIES

In this section, we conduct more ablation studies on the results in Section 5.1. Specifically, we focus on the best configuration in Table 3: Fast-LS-$l_0$ (T) (i.e., 1-step sTRADES (T) + SAT & N-FGSM). Unless specified, we adopt the same training settings as in Table 3.

Table 11 presents a performance comparison of the model when SAT is enable in different training phases. We can see that the performance achieves the best when enabling SAT at the 50-th epoch. This observation demonstrates that the best performance in 1-step sTRADES is achieved when SAT is enabled at the intermediate epoch where the learning rate is relatively low.

In Table 12, we compare the performance when using different labels, either the hard label from ground truth or the soft label by SAT, to generate adversarial perturbations for training. The results indicate that using soft labels to generate adversarial perturbations results in slightly better performance compared to using hard ones.

In Table 13, we compare the performance when using different momentum factor in SAT. We can see that the default setting in Huang et al. (2020), i.e., 0.9, provides the best performance.

In Table 14, we compare the performance when using different balance factor $\beta$ in TRADES. It can be observed that $\beta = 3$ and $6$ induce similar results, indicating the default setting in (Zhang et al., 2019b), i.e., 6, is the optimal.

Table 11: Ablation study on the epoch of enabling SAT. The evaluated attack is sAA, where the sparsity level $\epsilon = 20$.

| SAT epoch | 30 | 50 | 70 |
|---|---|---|---|
| Robust Accuracy | 60.2 | **63.0** | 62.8 |

Table 12: Ablation study on the labels used to generate adversarial samples. The evaluated attack is sAA, where the sparsity level $\epsilon = 20$.

| Label | Hard | Soft |
|---|---|---|
| Robust Accuracy | 62.6 | **63.0** |

Table 13: Ablation study on the momentum factor of SAT. The evaluated attack is sAA, where the sparsity level $\epsilon = 20$.

| SAT momentum | 0.5 | 0.7 | 0.9 |
|---|---|---|---|
| Robust Accuracy | 55.4 | 60.4 | **63.0** |

Table 14: Ablation study on the balance factor $\beta$ in TRADES loss function. The evaluated attack is sAA, where the sparsity level $\epsilon = 20$.

| TRADES $\beta$ | 1 | 3 | 6 |
|---|---|---|---|
| Robust Accuracy | 58.7 | **63.0** | **63.0** |

# F    IMPLEMENTATION DETAILS

Generally, the epoch of enabling SAT is $1/2$ of the total epochs. For N-FGSM, the random noise for augmentation is the random sparse perturbation with sparsity level ranging from 0 to $2\epsilon$, where $\epsilon$ is the sparsity level of adversarial perturbations. The interpolation factor $\alpha$ in trade-off loss function is set to 0.75. The balance factor $\beta$ in TRADES loss function is set to 6. The optimizer is SGD with a momentum factor of 0.9 and a weight decay factor of $5 \times 10^{-4}$. The learning rate is initialized to 0.05 and is divided by a factor of 10 at the $1/4$ and $3/4$ of the total epochs. The specific settings for different datasets are listed as follows:

- **CIFAR-10, CIFAR-100 (Krizhevsky et al., 2009) and GTSRB (Stallkamp et al., 2012):** The adopted network is PreAct ResNet-18 (He et al., 2016b) with softplus activation (Dugas et al., 2000). The training batch size is 128. We train the model for 100 epochs.
- **ImageNet-100 (Deng et al., 2009):** The adopted network is ResNet-34 (He et al., 2016a). The training batch size is 48. We train the model for 50 epochs.

Unless specified, the hyperparameters of attacks and other configurations are the same as in (Zhong et al., 2024).

