# OpenReview forum: "Fast Adversarial Training against Sparse Attacks Requires Loss Smoothing"
_ICLR.cc/2025/Conference — Submitted to ICLR 2025_

### Official Review · Reviewer_vSUq · 2024-10-18

**Soundness:** 3
**Presentation:** 3
**Contribution:** 3
**Rating:** 6
**Confidence:** 4

**Summary:**

This paper explores the challenges and solutions for catastrophic overfitting within 1-step adversarial training targeted at sparse adversarial perturbations bounded by the  $L_0$  norm.
Through a comprehensive analysis, the authors illustrate that the CO issue in $L_0$ norm stems from a more craggy loss landscape compared to its $L_{\infty}$,  $L_2$, and  $L_1$ counterparts.
To address these challenges, the authors propose incorporating soft labels and a trade-off loss function, strategies specifically designed to smooth the adversarial loss landscape associated with $ L_0$  norm 1-step adversarial training.

**Strengths:**

This work is the first to investigate fast adversarial training in the context of $L_0$ bounded perturbations.

The authors successfully demonstrate that the CO issue in the $L_0$ norm is caused by sub-optimal perturbation locations,  rather than sub-optimal perturbation magnitudes via some interesting ablation studies.

This study conducts extensive experiments, including ImageNet and Transformer-based architectures.

**Weaknesses:**

Theoretical analysis indicates that large $\left\|\boldsymbol{\delta}_1-\boldsymbol{\delta}_2\right\|$ can intensify the gradient discontinuity, and  $L_0$ norm has the largest upper bound.
However, directly comparing these upper bounds may not be a fair comparison due to the naturally larger freedom in change magnitudes associated with the $L_0$  norm.
Could authors provide some empirical results of $\left\|\boldsymbol{\delta}_1-\boldsymbol{\delta}_2\right\|$ among different norms to support their claim?

It would be interesting to examine the performance of Expectation over Transformation (EOT) [1] and Backward Pass Differentiable Approximation (BPDA) [1] on the authors’ proposed method, which can effectively exclude gradient masking in 1-step adversarial training.

Recent advances in addressing CO by preventing loss landscape distortion with other norms [2,3,4,5] may also prove beneficial in alleviating CO in the $L_0$ norm and should be discussed.

[1] Athalye, A., Carlini, N., & Wagner, D. (2018, July). Obfuscated gradients give a false sense of security: Circumventing defenses to adversarial examples. In International conference on machine learning (pp. 274-283). PMLR.\
[2] Grabinski, J., Jung, S., Keuper, J., & Keuper, M. (2022, October). Frequencylowcut pooling-plug and play against catastrophic overfitting. In European Conference on Computer Vision (pp. 36-57). Cham: Springer Nature Switzerland.\
[3] Lin, R., Yu, C., & Liu, T. (2024). Eliminating catastrophic overfitting via abnormal adversarial examples regularization. Advances in Neural Information Processing Systems, 36.\
[4] Rocamora, E. A., Liu, F., Chrysos, G., Olmos, P. M., & Cevher, V. Efficient local linearity regularization to overcome catastrophic overfitting. In The Twelfth International Conference on Learning Representations.\
[5] Lin, R., Yu, C., Han, B., Su, H., & Liu, T. Layer-Aware Analysis of Catastrophic Overfitting: Revealing the Pseudo-Robust Shortcut Dependency. In Forty-first International Conference on Machine Learning.

**Questions:**

Refer to weakness.

---

> ### Author Response · Authors · 2024-11-18
> **Rebuttal**
>
> We appreciate your constructive comments. We provide the responses to your concerns in a point-to-point manner.
>
> **1. Could authors provide some empirical results of |delta1-delta2| among different norms to support their claim?**
>
> We showcase the average $l_2$ distance between perturbations generated by 1-step and multi-step attacks $\|\delta_{one}-\delta_{multi}\|$ among different norms in the table below. The setting is the same as that in Table 5. It can be observed that **$\|\delta_{one}-\delta_{multi}\|$ of $l_0$ attacks is significantly larger than that of other attacks.** This is consistent with the theoretical conclusion.
>
> In addition, we showcased the average distances between gradients induced by 1-step and multi-step attacks in Table 5 in our manuscript. As presented in Table 5, the **average distance between gradients induced by 1-step and multi-step $l_0$ attacks is 5 orders of magnitude greater than those in other cases, even when a single pixel is perturbed.** This further demonstrates that the optimization is challenging in the context of $l_0$ adversarial training.
> | | $l_0~(\epsilon=20)$ | $l_1~(\epsilon=12)$ | $l_2~(\epsilon=0.5)$ | $l_\infty~(\epsilon=8/255)$ |
> |----|----|----|----|----|
> |$\|\|\delta_{one}-\delta_{multi}\|\|_2$ | **6.6635** | 1.5421 | 0.4901 | 1.4568|
>
> **2. It would be interesting to examine the performance of Expectation over Transformation (EOT) [1] and Backward Pass Differentiable Approximation (BPDA) [1].**
>
> We would like to recall that **our method is built on the adversarial training framework, which is the only valid method studied in [1] to withstand adaptive attacks.**
> As for the two mentioned methods, EOT is not expected to work since we do not randomly transform the input in evaluation, and BPDA is not applicable since non-differentiable operations, like purification and transformation, are not used in our models.
> Moreover, **Sparse-AutoAttack**, which is adopted in evaluation, **is an ensemble of both white-box and black-box attacks, where gradient masking can be evaded.**
>
> **3. Recent advances in addressing CO by preventing loss landscape distortion with other norms [2,3,4,5] may also prove beneficial in alleviating CO in the l0 norm and should be discussed.**
>
> We report the results of [2,3,5] in the table below. Note that **we exclude the results of [4] since it utilizes interpolation operation, which assumes the adversarial budget is a convex set, while the $l_0$ adversarial budget is non-convex.** The results indicate [2,3,5] experience catastrophic overfitting, thereby invalid in the $l_0$ case.
>
> | Model | FLC Pool [2] | N-AAER [3] | N-LAP [5] | Fast-LS-$l_0$|
> |-----|-----|-----|-----|-----|
> |sAA ($\epsilon=20$) |  *0.0* | *0.1* | *0.0* | **63.0**|
>
>
> **Reference**
>
> [1] Athalye, A., Carlini, N., & Wagner, D. (2018, July). Obfuscated gradients give a false sense of security: Circumventing defenses to adversarial examples. In International conference on machine learning (pp. 274-283). PMLR.
>
> [2] Grabinski, J., Jung, S., Keuper, J., & Keuper, M. (2022, October). Frequencylowcut pooling-plug and play against catastrophic overfitting. In European Conference on Computer Vision (pp. 36-57). Cham: Springer Nature Switzerland.
>
> [3] Lin, R., Yu, C., & Liu, T. (2024). Eliminating catastrophic overfitting via abnormal adversarial examples regularization. Advances in Neural Information Processing Systems, 36.
>
> [4] Rocamora, E. A., Liu, F., Chrysos, G., Olmos, P. M., & Cevher, V. Efficient local linearity regularization to overcome catastrophic overfitting. In The Twelfth International Conference on Learning Representations.
>
> [5] Lin, R., Yu, C., Han, B., Su, H., & Liu, T. Layer-Aware Analysis of Catastrophic Overfitting: Revealing the Pseudo-Robust Shortcut Dependency. In Forty-first International Conference on Machine Learning.

---

> > ### Comment · Reviewer_vSUq · 2024-11-22
> >
> > Thank you for the rebuttal. While most of my concerns are addressed, one question remains. Given that most CO defense methods are sensitive to hyperparameters and none of them were originally designed for $L_0$, did the authors conduct a hyperparameter search for baselines?

---

> ### Author Response · Authors · 2024-11-25
>
> We are glad that most of your concerns have been addressed. Regarding your remaining question, due to the time limit, we cannot conduct a hyperparameter search for all baselines in Table 6. Nevertheless, we test different hyperparameters of two representative baselines, GA [1] and NuAT [2]. Specifically, GA is the CO defense without soft labels and trade-off loss function, and NuAT performs the best among evaluated baselines.
>
> The results below indicate that **(1)** different $\lambda$ does not make GA applicable in fast $l_0$ adversarial training; **(2)** NuAT is sensitive to different hyperparameter choices as the training becomes unstable when $\lambda \geq 6$, and still underperforms our method, which achieves a robust accuracy of 63.0%.
>
> In summary, this ablation study further demonstrates the effectiveness of the proposed method. If our responses address your concerns, we would be grateful if you could adjust your scores.
>
> |$\lambda$ of GA |1|3|5|7|
> |---|---|---|---|---|
> |sAA ($\epsilon=20$)|0.2|0.0|0.0|0.0|
>
> |$\lambda$ of NuAT |1|3|6|8|
> |---|---|---|---|---|
> |sAA ($\epsilon=20$)|40.4|51.9|10.9|10.3|
>
> Reference
>
> [1] Andriushchenko et al., Understanding and improving fast adversarial training. NeurIPS 2020.
>
> [2] Sriramanan et al, Towards efficient and effective adversarial training. NeurIPS 2021.

---

> > ### Comment · Reviewer_vSUq · 2024-11-27
> >
> > Thank you for your response, which has addressed part of my major concerns, and I have decided to raise my score to 6. Considering the discussion period extension, I highly encourage the authors to conduct a hyperparameter search for all baselines in Table 6 to ensure a fair comparison.

---

> > > ### Author Response · Authors · 2024-11-28
> > >
> > > Thanks for your positive feedback. A comprehensive search of the hyperparameters for all baselines presented in Table 6 has been conducted. It should be noted that the invalid baselines, including **ATTA, GA, Fast-BAT, FLC Pooling, N-AAER, N-LAP, and LS**, remain ineffective after a comprehensive hyperparameter search. Nevertheless, we report the results for the remaining baselines, i.e., **AdvLC, MART, and Ours+AWP**,  in the table below.
> > >
> > > It can be observed that the performance of AdvLC significantly improves after hyperparameter search. However, it still underperforms our method, which achieves a robust accuracy of 63.0%. Furthermore, our method also benefits from AWP. However, AWP introduces additional computational overhead, thereby not being adopted in our method.
> > >
> > > |$\lambda$ of AdvLC |0.1|0.3|0.5|0.7|0.9|
> > > |---|---|---|---|---|---|
> > > |sAA ($\epsilon=20$)|9.3|47.6|57.1|59.6|10.3|
> > >
> > > |$\lambda$ of MART |1|3|5|7|
> > > |---|---|---|---|---|
> > > |sAA ($\epsilon=20$)|25.8|38.0|48.0|47.0|
> > >
> > > |$\gamma$ of Ours+AWP |0|1e-4|1e-3|1e-2|5e-2|
> > > |---|---|---|---|---|---|
> > > |sAA ($\epsilon=20$)|63.0|65.2|64.5|63.1|47.0|
> > >
> > > We would like to thank you again for your constructive comments that improve the quality of this work. We will update the results in a future version.

---

### Official Review · Reviewer_wG55 · 2024-10-29

**Soundness:** 3
**Presentation:** 3
**Contribution:** 2
**Rating:** 6
**Confidence:** 4

**Summary:**

This work studies the fast adversarial training, usually the single-step adversarial training (AT), in the setting of $\ell_0$-bounded sparse adversarial perturbation. It first identifies the phenomenon of catastrophic overfitting in $\ell_0$ fast AT and the accounts for it, sub-optimal perturbation locations and non-smoothness of the loss landscape. It then proposes to incorporate soft label and trade-off loss function into $\ell_0$ fast AT to smooth loss landscape to mitigate catastrophic overfitting.

**Strengths:**

1. The presentation is easy to follow
2. The proposed method is well-motivated.
3. The experimental results verify the effectiveness of the proposed method in mitigating catastrophic overfitting in $\ell_0$ fast AT.

**Weaknesses:**

1. The studied problem, catastrophic overfitting in $\ell_0$ fast AT, is quite narrow. It considers a very specific case, $\ell_0$, of the $\ell_p$ adversarial setting. The impact of this work's conclusions on the entire field, adversarial machine learning, is therefore limited.
2. The analytical framework and the findings are similar to the existing works of analyzing overfitting in AT. Some of them are already cited in this work, while some else are missing. For example, [1] also attributes overfitting in AT to the degradation of training adversary and the complication of loss landscape throughout training, and proposes to integrate loss landscape smooth techniques into AT to mitigate overfitting. However, I acknowledge that some conlcusion of this work, like the one about sub-optimal perturbation location, seems to be unique to the setting of $\ell_0$ and useful.
3. The mitigation methods, soft label and trade-off loss function, are proposed by the exiting works, so the methodological contribution of this work is the discovery of the effectiveness of existing methods in mitigating catastrophic overfitting in $\ell_0$ fast AT, but not something novel like proposing a new method. However, as mentioned in Weakness 2, the idea of mitigating overfitting in AT through loss smoothing has been already explored in the existing works, despite that different smoothing techniques are employed.

[1] Li, Lin, and Michael Spratling. "Understanding and combating robust overfitting via input loss landscape analysis and regularization." Pattern Recognition 136 (2023): 109229.

**Questions:**

Please find above.

---

> ### Author Response · Authors · 2024-11-18
> **Rebuttal (1/2)**
>
> We appreciate your constructive comments. We provide the responses to your concerns in a point-to-point manner.
>
> **1. The studied problem, catastrophic overfitting in $l_0$ fast AT, is quite narrow.**
>
> We would like to highlight that **(1)** the non-convexity of $l_0$ adversarial budget and the craggy loss landscape make fast $l_0$ adversarial training more challenging than other cases; **(2)** Various existing methods to mitigate catastrophic overfitting do not work in the $l_0$ case; **(3)** Sparse perturbations are common in physical work, so investigating robustness against sparse perturbations is important. **(4)** Fast $l_0$ adversarial training solves a discrete optimization problem, which is fundamentally different from the convex and continuous constraints in other $l_p$ norm cases, our methods can help other problems in this field that involve discrete optimization as well.
>
> In the following, we elaborate on the aforementioned viewpoints:
>
> 1. **Fast $l_0$ adversarial training is more challenging than the $l_1$, $l_2$ and $l_\infty$ cases**. To substantiate this claim, we analyze the phenomenon of **catastrophic overfitting** in fast $l_0$​ adversarial training in Section 3, concluding that **it arises from suboptimal perturbation locations and a significantly more challenging loss landscape compared to other norms**.
> 2. The results in Table 6 reveal that **the existing fast adversarial training methods against $l_1$, $l_2$, $l_\infty$ attacks, like random start [1], GradAlign [2], and ATTA [3], cannot avoid catastrophic overfitting in the $l_0$ scenario.** Therefore, it is necessary to develop an effective fast adversarial training approach against sparse attacks to fill the void.
> 3. In contrast to $l_\infty$ or $l_2$ bounded perturbations, $l_0$ bounded perturbations are sparse and quite common in physical scenarios, including broken pixels in LED screens to fool object detection models and adversarial stickers on road signs to make an auto-driving system fail [6-9]. In this regard, **training a robust model against l0 bounded perturbations is essential to realistic scenarios with high-security needs, such as auto-driving system and access control system.**
> 4. Due to the unique properties of the $l_0$, we are solving an optimization problem with discrete constraints in this work. This is fundamentally different from other $l_p$ norm cases, which are continuous and convex. **In this regard, the analyses and the methods in our work can help design efficient algorithms to solve other problems in this field which involve discrete optimization**, such as adversarial perturbations in texts [10] and quantized parameters [11].
>
> **2. The analytical framework and the findings are similar to the existing works of analyzing overfitting in AT.**
>
> We acknowledge the relevance of AdvLC to this work and will cite it in the revised version. However, we find:
>
> 1. **Although the degraded attack effectiveness can be mitigated by multi-step attacks, Figure 3 illustrates that the craggy loss landscape in $l_0$ adversarial training can still lead to catastrophic overfitting (CO).** In contrast, AdvLC attributes CO to degraded perturbations and claims that the craggy loss landscape aggravates the degradation.
> 2.  **We propose theoretically guaranteed methods to improve both Lipschitz continuity and Lipschitz smoothness.** However, AdvLC penalizes logit difference to regularize gradient to improve attack effectiveness, which is similar to the effect of soft labels and can only improve Lipschitz continuity.
> 3. **We also employ AdvLC in fast $l_0$ adversarial training and obtain a robust accuracy of 47.6% against Sparse-AutoAttack, while our method achieves a robust accuracy of 63.0%.** This indicates that only improving Lipschitz continuity is insufficient in the $l_0$ case, underscoring the requirement of more comprehensive smoothing techniques for $l_0$ adversarial training.

---

> ### Author Response · Authors · 2024-11-18
> **Rebuttal (2/2)**
>
> **3. The mitigation methods, soft label and trade-off loss function, are proposed by the exiting works, so the methodological contribution of this work is not something novel like proposing a new method.**
>
> The differences between our work and AdvLC are elaborated in the response to the second comment.
>
> As for the novelty, We would like to emphasize that the rationale and advantages of employing soft labels and the trade-off loss function differ significantly in the context of $l_0$ norms. **Specifically, the use of soft labels and trade-off loss functions is intended to improve generalization and to balance the robustness-utility trade-off in scenarios involving other norms such as $l_1$​, $l_2$, and $l_\infty$​. However, these techniques are crucial for achieving meaningful performance in fast adversarial training under $l_0$​ constraints.** Without them, fast adversarial training against $l_0$​-bounded perturbations usually suffer from catastrophic overfitting, leading to only trivial performance outcomes (see Fig. 1). In contrast, fast adversarial training using other norms can still achieve competitive results through various techniques, such as random start [1], GradAlign [2], and ATTA [3]. These methods, however, are ineffective in the $l_0$​ scenario (see Table 6). In summary, extensive analyses and experiments indicate that these smoothing techniques are the key to solving catastrophic overfitting in $l_0$ adversarial training, which is not the case for other norms.
>
> To substantiate our claims, we analyze the phenomenon of **catastrophic overfitting** in fast $l_0$​ adversarial training in Section 3, concluding that **it arises from suboptimal perturbation locations and a significantly more challenging loss landscape compared to other norms**. In Section 4, we propose that **soft labels and the trade-off loss function can provably enhance first- and second-order smoothness, respectively**. Numerical results support this assertion (see Fig. 6).
>
> In Section 5.1, we evaluate various combinations of methods that incorporate soft labels and the trade-off loss function. Our findings indicate that the integration of TRADES, SAT, and N-FGSM achieves the highest performance. **Notably, our method attains the best robust accuracy of 63.0%**, whereas other smoothing techniques, such as NuAT [4] and MART [5], report robust accuracies of only 51.9% and 48.0%, respectively (see Table 6).
>
> In summary, our work is definitely not a simple extension of existing techniques to a related task. Instead, we exploit the unique challenges and solutions to a new problem not explored before. Our proposed methods borrow some elements from existing methods, but the rationale and benefits of employing them are different. Extensive experiments demonstrate that our method can reliably solve the challenge and achieve state-of-the-art performance.
>
> **Reference**
>
> [1] Wong et al., Fast is better than free: Revisiting adversarial training. ICLR 2020.
>
> [2] Andriushchenko et al., Understanding and improving fast adversarial training. NeurIPS 2020.
>
> [3] Zheng et al., Efficient Adversarial Training with Transferable Adversarial Examples. CVPR 2020.
>
> [4] Sriramanan et a.l, Towards efficient and effective adversarial training. NeurIPS 2021.
>
> [5] Wang et al., Improving adversarial robustness requires revisiting misclassified examples. ICLR 2020.
>
> [6] Papernot et al., Practical black-box attacks against machine learning. ACM ASIACCS 2017
>
> [7] Akhtar et al., Threat of adversarial attacks on deep learning in computer vision: A survey. IEEE Access 2018.
>
> [8] Xu et al., Adversarial attacks and defenses in images, graphs and text: A review. International Journal of Automation and Computing 2019.
>
> [9] Feng et al., Graphite: Generating automatic physical examples for machine-learning attacks on computer vision systems. EuroS&P 2022.
>
> [10] Yuan, Lifan, et al. "Bridge the Gap Between CV and NLP! A Gradient-based Textual Adversarial Attack Framework." Findings of the Association for Computational Linguistics: ACL 2023. 2023.
>
> [11] Yang, Yulong, et al. "Quantization aware attack: Enhancing transferable adversarial attacks by model quantization." IEEE Transactions on Information Forensics and Security (2024).

---

> > ### Comment · Reviewer_wG55 · 2024-11-21
> >
> > Dear Authors,
> >
> > Thank you for your detailed responses to my concerns and for providing additional experimental results. Nevertheless, after carefully reviewing your responses, I am sorry to say that some of my concerns remain unresolved:
> >
> > 1. While I recognize the importance of defending against $l_0$-bounded adversarial examples, I find it challenging to identify a broader takeaway from the analysis and methods proposed in this work that could extend beyond $l_0$ defense. This is why the scope feels narrow. Although the authors suggest that the proposed methods may inspire future research on discrete optimization in this domain, this claim is not immediately evident and would require further supporting evidence.
> >
> > 2. Regarding Weakness 3, the authors' response has not changed my perspective on the limited technical novelty of the proposed methods. Both the use of soft labels and the trade-off loss have been introduced in prior works, irrespective of how they perform under different $l_p$ norms. Furthermore, the argument that "these techniques are more crucial for $l_0$" does not, in my view, substantiate the novelty of the proposed methods.

---

> ### Author Response · Authors · 2024-11-25
>
> Thanks for your constructive comments. We provide further responses to your remaining concerns:
>
> **1. Broader impact**
>
> To explore the broader impact of our method, we employ our method in adversarial training on tabular data. We aim to perturb **one input feature** of DANet [1] trained on the forest cover type dataset (54 features in total). The training attack is 1-step sPGD ($\epsilon_{train} = 6$) and the evaluation attack is 10000-step sPGD ($\epsilon = 1$). Note that we adapt sPGD to accommodate tabular data. The results indicate that our method is still effective for tabular data. Due to the time limit, we cannot explore more application scenarios. Despite that, the results in Table 4, 7,8,10, and below can already demonstrate the effectiveness of our method across different datasets, networks, and modalities.
>
> |Model|Vanilla|1-step AT|Ours|
> |---|---|---|---|
> |Clean Acc.|93.9|88.5|86.4|
> |Robust Acc.|0.2|31.5|**45.8**|
>
> **2. Novelty**
>
> It is important to note that **the rationale and benefits of the adopted methods are distinct from their original purpose**. Specifically, SAT was proposed to improve the generalization across various tasks, while TRADES was initially designed to improve the robustness-accuracy trade-off. However, **they turn out indispensable in mitigating catastrophic overfitting in fast $l_0$ adversarial training.** More details are illustrated in the first point of our general response.
>
> Similarly, the differences between FGSM [2], RS-FGSM [3] and N-FGSM [4] are that RS-FGSM adopts random initialization, which has been widely adopted in almost all aspects of deep learning; N-FGSM augments clean samples with random noise, which is generally used to improve generalization. It should be noted that **neither RS-FGSM nor N-FGSM introduces any “novel” approach. However, both of them do have profound implications for fast adversarial training.**
>
> Reference
>
> [1] Chen et al. DANETs: Deep Abstract Networks for Tabular Data Classification and Regression. AAAI 2022.
>
> [2] Goodfellow et al. Explaining and harnessing adversarial examples. International Conference on Learning Representations (ICLR), 2015.
>
> [3] Wong et al. Fast is better than free: Revisiting adversarial training. In International Conference on Learning Representations (ICLR), 2020.
>
> [4] Jorge et al. Make Some Noise: Reliable and Efficient Single-Step Adversarial Training. NeurIPS 2022.

---

> > ### Author Response · Authors · 2024-12-02
> > **Looking Forward to the Reviewer's Reply**
> >
> > Dear reviewer wG55,
> >
> > We thank again for your detailed comments and constructive suggestions. As the deadline approaches, we kindly ask if our responses have adequately answered your questions to address your concerns. We would greatly appreciate your feedback and discussion with you to ensure that we have fully resolved the outstanding issues for the improvement of this manuscript. Thank you for your time, effort and consideration.

---

> > ### Comment · Reviewer_wG55 · 2024-12-03
> >
> > Hi,
> >
> > Thank you for your detailed responses. I appreciate the new application scenario demonstrated by the authors and partially agree with the statement that novelty is not always necessary for impactful work. However, these points do not directly address my core concerns:
> >
> > 1. **Practical Impact:** It remains unclear how the identified phenomenon and proposed method can benefit fast adversarial training beyond sparse settings.
> > 2. **Technical Novelty:** While the authors highlight "different rationales and benefits," this does not increase the technical novelty. These techniques are fundamentally similar to those proposed in prior work.
> >
> > Overall, I appreciate the authors' efforts in addressing my concerns, especially the new experimental results. I acknowledge that some of my concerns have been addressed. Therefore, I have decided to raise my score to 6, indicating a borderline accept. The reasons for not assigning a higher score are outlined above.

---

> > > ### Author Response · Authors · 2024-12-04
> > >
> > > Dear reviewer wG55,
> > >
> > > We are grateful for your positive feedback and truly appreciate your insightful comments during the rebuttal. We believe the constructive discussion helped us improve the quality of this work. Thank you once again for your time and consideration.

---

### Official Review · Reviewer_Jqqa · 2024-11-02

**Soundness:** 3
**Presentation:** 3
**Contribution:** 2
**Rating:** 6
**Confidence:** 4

**Summary:**

This paper investigates FAT against sparse adversarial perturbations bounded by the $L_0$ norm, highlighting the challenges associated with FAT, including performance degradation and catastrophic overfitting phenomenon. The authors propose Fast-LS-$l_0$, which incorporates soft labels and a trade-off loss function to smooth the adversarial loss landscape. Experimental results show the effectiveness of the proposed method against sparse attacks.

**Strengths:**

1.	Research on FAT under $L_0$ norm constraints is relatively limited, and the authors address this gap by proposing an innovative method to mitigate performance degradation and catastrophic overfitting in FAT.

2.	The authors provide detailed theoretical insights into the connection between the non-smooth adversarial loss landscape and catastrophic overfitting, giving clarity to the existing issues from the theoretical perspective.

3.	The paper is well-organized and clearly explained, and extensive experiments across diverse datasets validate the robustness of the proposed method.

**Weaknesses:**

1.	The novelty is somewhat limited, as many techniques, including soft labels and TRADES loss, are widely used in adversarial defense. The overall method may appear as a straightforward integration of existing techniques.
2.	The application scenario is not clearly defined, particularly why robustness under the $L_0$ norm constraint is essential. Can this adversarial training effectively enhance defense against one-pixel attacks?
3.	The application scenario is unclear, and it is unclear why the model needs to be more robust under the $L_0$ norm constraint. Can this adversarial training enhance the defense capability under the one-pixel attack?
4.	Whilemost current methods are based on $L_{\infty}$ norm constraints, the author should show the performance of these methods under $L_0$ constraints, such as FAST-GA[1], FAST-BAT[2], FAST-PCO[3].

[1]. Andriushchenko, M. et.al “Understanding and improving fast adversarial training” NIPS, 2020
[2]. Zhang Y. et.al “Revisiting and advancing fast adversarial training through the lens of bi-level optimization” PMLR, 2022
[3]. Wang Z. et.al “Preventing Catastrophic Overfitting in Fast Adversarial Training: A Bi-level Optimization Perspective” ECCV, 2024

**Questions:**

1.	Can Fast-LS-$l_0$ be adapted for other sparse attacks, such as those bounded by $L_1$?
2.	Does the trade-off loss function parameter $\alpha$ need to be adjusted across different datasets and tasks?
3.	Could you further elaborate on the necessity of adversarial training under the $L_0$ constraint?

---

> ### Author Response · Authors · 2024-11-18
> **Rebuttal (1/2)**
>
> We appreciate your constructive comments. We provide the responses to your concerns in a point-to-point manner.
>
> **1. The novelty is somewhat limited. The overall method may appear as a straightforward integration of existing techniques.**
>
> We would like to emphasize that the rationale and advantages of employing soft labels and the trade-off loss function differ significantly in the context of $l_0$ norms. **Specifically, the use of soft labels and trade-off loss functions is intended to improve generalization and to balance the robustness-utility trade-off in scenarios involving other norms such as $l_1$​, $l_2$, and $l_\infty$​. However, these techniques are crucial for achieving meaningful performance in fast adversarial training under $l_0$​ constraints.** Without them, fast adversarial training against $l_0$​-bounded perturbations usually suffer from catastrophic overfitting, leading to only trivial performance outcomes (see Fig. 1). In contrast, fast adversarial training using other norms can still achieve competitive results through various techniques, such as random start [1], GradAlign [2], and ATTA [3]. These methods, however, are ineffective in the $l_0$​ scenario (see Table 6).
>
> To substantiate our claims, we analyze the phenomenon of **catastrophic overfitting** in fast $l_0$​ adversarial training in Section 3, concluding that **it arises from suboptimal perturbation locations and a significantly more complex loss landscape compared to other norms**. In Section 4, we propose that **soft labels and the trade-off loss function can provably enhance first- and second-order smoothness, respectively**. Numerical results support this assertion (see Fig. 6).
>
> Motivated by these analyses and observations, in Section 5.1, we evaluate various combinations of methods that incorporate soft labels and the trade-off loss function. Our findings indicate that the integration of TRADES, SAT, and N-FGSM achieves the highest performance. **Notably, our method attains the best robust accuracy of 63.0%**, whereas other smoothing techniques, such as NuAT [4] and MART [5], report robust accuracies of only 51.9% and 48.0%, respectively (see Table 6).
>
> In summary, our work is definitely not a simple extension of existing techniques to a related task. Instead, we exploit the unique challenges and solutions to a new problem not explored before. Our proposed methods borrow some elements from existing methods, but the rationale and benefits of employing them are different. Extensive experiments demonstrate that our method can reliably solve the challenge and achieve state-of-the-art performance.
>
> **2. The application scenario is not clearly defined. Can this adversarial training effectively enhance defense against one-pixel attacks?**
>
> In contrast to $l_\infty$ or $l_2$ bounded perturbations, $l_0$ bounded perturbations are sparse and quite common in physical scenarios, including broken pixels in LED screens to fool object detection models and adversarial stickers on road signs to make an auto-driving system fail [6-10]. In this regard, **training a robust model against l0 bounded perturbations is essential to realistic scenarios with high-security needs, such as auto-driving system and access control system.**
>
> Although $l_0$ bounded perturbations have been studied in many existing works [6-10], most of them focus on generating adversarial perturbations instead of the defense against these attacks. Moreover, fast adversarial training against l0 bounded perturbations is challenging due to the suboptimality of one-step attacks and the craggy loss landscape induced by l0 bounded perturbations. **What’s worse, the existing fast adversarial training methods against $l_1$, $l_2$, $l_\infty$ attacks, like random start [1], GradAlign [2], and ATTA [3], cannot avoid catastrophic overfitting in the $l_0$ scenario.** Therefore, it is necessary to develop an effective fast adversarial training approach against sparse attacks to fill the void.
>
> Additionally, our method can enhance defense against one-pixel attack by setting $\epsilon=1$. The results are reported in the table below, where vanilla denotes the model without adversarial training. The robust accuracy against one-pixel attack [15] increases from 79.9% to 81.5%.
> |  | Clean Acc | One-Pixel | RS | sPGD$_{proj}$ | sPGD$_{unproj}$  | sAA|
> |-----|-----|-----|-----|-----|-----|-----|
> |Vanilla | 94.2 | 79.9 | 79.5 | 80.6 | 79.9 | 78.7|
> |Fast-LS-$l_0$| **82.5** | **81.5**|**81.7**|**81.1**|**81.4**|**81.1**|

---

> ### Author Response · Authors · 2024-11-18
> **Rebuttal (2/2)**
>
> **3. The author should show the performance of these methods under l0 constraints, such as FAST-GA, FAST-BAT, FAST-PCO.**
>
> We report the results of FAST-GA and FAST-BAT below. Specifically, **Fast-GA** was already evaluated in our manuscript (see GA in Table 6) and **was shown invalid in the $l_0$ case**, the model still suffers from catastrophic overfitting. Additionally, **Fast-BAT turns out unstable and performs poorly in the $l_0$ case**. Note that the result of **FAST-PCO is not included since it was designed for $l_\infty$ attack and utilizes interpolation, which assumes the adversarial budget is a convex set, while the $l_0$ adversarial budget is non-convex.**
> | |Fast-GA | Fast-BAT | Fast-LS-$l_0$|
> |-----|-----|-----|-----|
> |sAA ($\epsilon=20$)| *0.0* | 14.1 | **63.0**|
>
> **4. Can Fast-LS-l0 be adapted for other sparse attacks, such as those bounded by $l_1$ ?**
>
> First, we would like to point out that **perturbations bounded by $l_1$ norm are not necessarily sparse [11].** Actually, [11] shows that models suffering from catastrophic overfitting in the $l_1$ norm case are usually not robust to non-sparse perturbations in the $l_1$ adversarial budget. Despite this, we still report the results on AutoAttack-$l_1$ [12].
>
> We need to emphasize that **Fast-LS-$l_0$ is not expected to be robust against $l_1$ attack since $l_1$ attack does not guarantee sparsity,** because **shreds of evidence suggest that models trained against specific perturbations are not robust to other perturbations [12, 13].** Nevertheless, Fast-LS-$l_0$ still significantly outperforms sTRADES, revealing its effectiveness in improving generalization among different attacks. Notably, efficient training algorithms exist to obtain competitive performance in robustness against $l_1$ bounded perturbations, such as Fast-EG-$l_1$ [11].
>
> |Model| sTRADES |Fast-LS-$l_0$|
> |-----|-----|-----|
> |AA-$l_1$ ($\epsilon=12$)| 0.2 | **14.1** |
>
> **5. Does the trade-off loss function parameter alpha need to be adjusted across different datasets and tasks?**
>
> The corresponding ablation study was already conducted, please see the results in Table 14 in our manuscript. Specifically, the optimal $\beta$ in TRADES (i.e., $\alpha$ in trade-off loss function) is 6, which is the same as the default one that is obtained based $l_\infty$ attack.  Empirically, this choice can generate competitive performance across different scenarios, which frees the practitioners of hyper-parameter tuning.
>
> **6.Could you further elaborate on the necessity of adversarial training under the l0 constraint?**
>
> It is necessary to investigate adversarial training under $l_0$ constraint. Adversarial training is shown as the most reliable method to achieve adversarial robustness in extensive case studies in [14]. Further details are in the response to the second comment.
>
> **Reference**
>
> [1] Wong et al., Fast is better than free: Revisiting adversarial training. ICLR 2020.
>
> [2] Andriushchenko et al., Understanding and improving fast adversarial training. NeurIPS 2020.
>
> [3] Zheng et al., Efficient Adversarial Training with Transferable Adversarial Examples. CVPR 2020.
>
> [4] Sriramanan et a.l, Towards efficient and effective adversarial training. NeurIPS 2021.
>
> [5] Wang et al., Improving adversarial robustness requires revisiting misclassified examples. ICLR 2020.
>
> [6] Papernot et al., Practical black-box attacks against machine learning. ACM ASIACCS 2017
>
> [7] Akhtar et al., Threat of adversarial attacks on deep learning in computer vision: A survey. IEEE Access 2018.
>
> [8] Xu et al., Adversarial attacks and defenses in images, graphs and text: A review. International Journal of Automation and Computing 2019.
> [9] Feng et al., Graphite: Generating automatic physical examples for machine-learning attacks on computer vision systems. EuroS&P 2022.
>
> [11] Jiang et al., Towards Stable and Efficient Adversarial Training against l1 Bounded Adversarial Attacks. ICML 2023.
>
> [10] Wei et al., Unified adversarial patch for cross-modal attacks in the physical world. ICCV
>
> [12] Croce et al., Mind the Box: l1-APGD for Sparse Adversarial Attacks on Image Classifiers. ICML 2021.
>
> [13] Zhong et al., Towards Efficient Training and Evaluation of Robust Models against l0 Bounded Adversarial Perturbations. ICML 2024.
>
> [14] Athalye et al. "Obfuscated gradients give a false sense of security: Circumventing defenses to adversarial examples." ICML 2018.
>
> [15] Tom Yuviler and Dana Drachsler-Cohen. 2023. One Pixel Adversarial Attacks via Sketched Programs. Proc. ACM Program. Lang. 7, PLDI, Article 187 (June 2023).

---

> > ### Comment · Reviewer_Jqqa · 2024-12-02
> >
> > Thanks for the author's reply and additional experiments. Similar concerns are included in other reviewers' responses, including limitations of technical innovations, even though the authors emphasized different reasons for adopting them. I will consider raising my score.

---

> > > ### Author Response · Authors · 2024-12-02
> > >
> > > Thanks for your reply. We appreciate your willingness to raise your score. However, we noticed that the score hasn't changed. Is there anything else we can help you with?

---

> ### Author Response · Authors · 2024-11-25
>
> It is a gentle reminder that the discussion period is coming to an end. We kindly request you to review our response to see if it sufficiently addresses your concerns. Your feedback is of great importance to us. For your convenience, we summarize some important points during the discussion with other reviewers here:
>
> 1. In the responses to reviewer wG55, we further evaluate our method on tabular data.  The results indicate that our method is still effective for tabular data. **The results in Table 4, 7,8,10, and there demonstrate the effectiveness of our method across different datasets, networks, and modalities.**
>
> 2. We further highlight that **the rationale and benefits of the adopted methods are distinct from their original purpose, and they turn out indispensable in mitigating catastrophic overfitting in fast $l_0$ adversarial training.** More details are illustrated in the second point of the responses to reviewer wG55.
>
> 3. Following the suggestion of reviewer vSUq, we conduct additional experiments to show that the poor performance of evaluated baselines is not attributed to the hyperparameter selection. This further indicates the effectiveness of our method.

---

### Author Response · Authors · 2024-11-18
**General Response (1/2)**

We would like to sincerely thank the reviewers and ACs for their engagement and constructive review.

We are very pleased that the reviewers recognized our work as “an innovative method to mitigate performance degradation and catastrophic overfitting in FAT under $\ell_0$ norm” (Jqqa) and "well-motivated" (wG55). We are also encouraged to see that all three reviewers found our proposed method effective and extensively validated across diverse datasets. Additionally, we appreciate the acknowledgment of our theoretical analysis, described as “detailed and clear” (Jqqa), as well as the recognition of our ablation studies for demonstrating the uniqueness of catastrophic overfitting under the $\ell_0$ norm from other norms as “interesting” (vSUq) and "useful" (wG55).

The following are our responses to the most common comments from reviewers. Please refer to the point-to-point responses to each reviewer for further details.

**1. Novelty**

We would like to emphasize that the rationale and advantages of employing soft labels and the trade-off loss function differ significantly in the context of $l_0$ norms. **Specifically, the use of soft labels and trade-off loss functions is intended to improve generalization and to balance the robustness-utility trade-off in scenarios involving other norms such as $l_1$​, $l_2$, and $l_\infty$​. However,  in terms of fast adversarial training under $l_0$​ constraints, these techniques are crucial for achieving meaningful performance, particularly without suffering from catastrophic overfitting.** Without them, fast adversarial training against $l_0$​-bounded perturbations usually suffer from catastrophic overfitting, leading to only trivial performance outcomes (see Fig. 1). In contrast, fast adversarial training using other norms can still achieve competitive results through various techniques, such as random start [1], GradAlign [2], and ATTA [3]. These methods, however, are ineffective in the $l_0$​ scenario (see Table 6).

To substantiate our claims, we analyze the phenomenon of **catastrophic overfitting** in fast $l_0$​ adversarial training in Section 3, concluding that **it arises from suboptimal perturbation locations and a significantly more challenging loss landscape compared to other norms**. In Section 4, we propose that **soft labels and the trade-off loss function can provably enhance first- and second-order smoothness, respectively**, which are crucial to mitigate catastrophic overfitting. Numerical results support this assertion (see Fig. 6).

Motivated by these analyses and observations, in Section 5.1, we evaluate various combinations of methods that incorporate soft labels and the trade-off loss function. Our findings indicate that the integration of TRADES, SAT, and N-FGSM achieves the highest performance. **Notably, our method attains the best robust accuracy of 63.0%**, whereas other smoothing techniques, such as NuAT [4] and MART [5], report robust accuracies of only 51.9% and 48.0%, respectively (see Table 6).

In summary, this work is first to investigate the **unique** challenges and solutions in accelerated adversarial training against $l_0$ bounded perturbations, **it achieves the state-of-the-art performance based on the most comprehensive robustness evaluation so far**.

**2. Application Scenarios**

In contrast to $l_\infty$ or $l_2$ bounded perturbations, $l_0$ bounded perturbations are sparse and quite common in physical scenarios, including broken pixels in LED screens to fool object detection models and adversarial stickers on road signs to make an auto-driving system fail [6-10]. In this regard, **training a robust model against $l_0$ bounded perturbations is essential to realistic scenarios with high-security needs, such as auto-driving systems and access control systems.**

Although $l_0$ bounded perturbations have been studied in many existing works [6-10], most of them focus on generating adversarial perturbations instead of the defense against these attacks. Moreover, fast adversarial training against $l_0$ bounded perturbations is challenging due to the suboptimality of one-step attacks and the craggy loss landscape induced by $l_0$ bounded perturbations. **What’s worse, the existing fast adversarial training methods against $l_1$, $l_2$, $l_\infty$ attacks, like random start [1], GradAlign [2], and ATTA [3], cannot avoid catastrophic overfitting and thus are invalid in the $l_0$ scenario.** Therefore, it is necessary to develop an effective fast adversarial training approach against sparse attacks to fill the gap.

---

> ### Author Response · Authors · 2024-11-18
> **General Response (2/2)**
>
> **3. Additional Experiments**
>
> To address the reviews’ concerns, we further compare our method with other methods to mitigate catastrophic overfitting. The results listed below suggest that our method achieves the strongest robustness against sparse attacks. More specific experimental results are included in the point-to-point responses.
> | Model | Fast-BAT [10] | AdvLC [11] | FLC Pool [12] | N-AAER [13] | N-LAP [14] | Fast-LS-$l_0$|
> |-----|-----|-----|-----|-----|-----|-----|
> |sAA ($\epsilon=20$) | 14.1 | 47.6 | *0.0* | *0.1* | *0.0* | **63.0**|
>
> **4. Revision to the Manuscript**
>
> We revise the manuscript as follows and highlight them in red.
>
> - Cite and discuss [10-14] in the manuscript.
> - Add results of suggested methods [10-14] to mitigate catastrophic overfitting in Table 6 of the manuscript.
>
> **Reference**
>
> [1] Wong et al., Fast is better than free: Revisiting adversarial training. ICLR 2020.
>
> [2] Andriushchenko et al., Understanding and improving fast adversarial training. NeurIPS 2020.
>
> [3] Zheng et al., Efficient Adversarial Training with Transferable Adversarial Examples. CVPR 2020.
>
> [4] Sriramanan et a.l, Towards efficient and effective adversarial training. NeurIPS 2021.
>
> [5] Wang et al., Improving adversarial robustness requires revisiting misclassified examples. ICLR 2020.
>
> [6] Papernot et al., Practical black-box attacks against machine learning. ACM ASIACCS 2017
>
> [7] Akhtar et al., Threat of adversarial attacks on deep learning in computer vision: A survey. IEEE Access 2018.
>
> [8] Xu et al., Adversarial attacks and defenses in images, graphs and text: A review. International Journal of Automation and Computing 2019.
>
> [9] Feng et al., Graphite: Generating automatic physical examples for machine-learning attacks on computer vision systems. EuroS&P 2022.
>
> [10] Zhang Y. et.al “Revisiting and advancing fast adversarial training through the lens of bi-level optimization” PMLR, 2022
>
> [11] Li, Lin, and Michael Spratling. "Understanding and combating robust overfitting via input loss landscape analysis and regularization." Pattern Recognition 136 (2023): 109229.
>
> [12] Grabinski, J., Jung, S., Keuper, J., & Keuper, M. (2022, October). Frequencylowcut pooling-plug and play against catastrophic overfitting. In European Conference on Computer Vision (pp. 36-57). Cham: Springer Nature Switzerland.
>
> [13] Lin, R., Yu, C., & Liu, T. (2024). Eliminating catastrophic overfitting via abnormal adversarial examples regularization. Advances in Neural Information Processing Systems, 36.
>
> [14] Lin, R., Yu, C., Han, B., Su, H., & Liu, T. Layer-Aware Analysis of Catastrophic Overfitting: Revealing the Pseudo-Robust Shortcut Dependency. In Forty-first International Conference on Machine Learning.

---

### Meta-Review · Area_Chair_6jAD · 2024-12-18

**Metareview:**

The paper addresses an important issue in fast adversarial training against sparse perturbations, providing a specific solution to mitigate catastrophic overfitting through loss smoothing techniques like soft labels and trade-off loss functions. However, the proposed approach lacks significant technical novelty, as it primarily builds on existing methods and combines them without introducing a fundamentally new framework. While the experimental results demonstrate effectiveness under sparse attacks, the paper’s scope is narrow, focusing exclusively on this specific setting without offering broader insights or applications to other adversarial scenarios. The theoretical claims about the challenging loss landscape are compelling but insufficiently supported by empirical evaluations, particularly under stronger adaptive attack scenarios such as BPDA or EoT. Furthermore, the real-world relevance of defending against sparse perturbations remains limited, and the comparison with baseline methods raises concerns about fairness and comprehensiveness despite subsequent hyperparameter tuning. Overall, the contribution does not meet the bar for acceptance due to its constrained impact and insufficient novelty.

**Additional Comments On Reviewer Discussion:**

During the rebuttal period, reviewers raised concerns about the technical novelty, narrow scope, limited empirical validation, baseline comparisons, and real-world impact of the submission. While the authors argued that their integration of soft labels and trade-off loss functions is crucial for mitigating catastrophic overfitting in sparse perturbations, the reliance on established methods limited the perceived innovation. The authors provided additional experiments, including evaluations on tabular data and hyperparameter tuning for baselines, which addressed some fairness concerns and improved comparisons. However, the lack of broader applicability beyond sparse perturbations, insufficient empirical evidence to validate theoretical claims under adaptive attacks like BPDA and EOT, and the limited demonstration of real-world impact reduced the work’s overall contribution. Although the authors made a strong case for the importance of addressing sparse perturbations, the incremental nature of the contribution and the constrained focus led to a decision not to recommend acceptance.

---

### Decision · Program_Chairs · 2025-01-22

Reject